# CALIBRATING UNCERTAINTY FOR ZERO-SHOT ADVERSARIAL CLIP

## ABSTRACT

CLIP delivers strong zero-shot classification but remains highly vulnerable to adversarial attacks. Previous work of adversarial fine-tuning largely focuses on matching the predicted logits between clean and adversarial examples, which overlooks uncertainty calibration and may degrade the zero-shot generalization. A common expectation in reliable uncertainty estimation is that predictive uncertainty should increase as inputs become more difficult or shift away from the training distribution. However, we frequently observe the opposite in the adversarial setting: perturbations not only degrade accuracy but also suppress uncertainty, leading to severe miscalibration and unreliable over-confidence. This overlooked phenomenon highlights a critical reliability gap beyond robustness. To bridge this gap, we propose a novel adversarial fine-tuning objective for CLIP considering both prediction accuracy and uncertainty alignments. By reparameterizing the output of CLIP as the concentration parameter of a Dirichlet distribution, we propose a unified representation that captures relative semantic structure and the magnitude of predictive confidence. Our objective aligns these distributions holistically under perturbations, moving beyond single-logit anchoring and restoring calibrated uncertainty. Experiments on multiple zero-shot classification benchmarks demonstrate that our approach effectively restores calibrated uncertainty and achieves competitive adversarial robustness while maintaining clean accuracy.

## 1 INTRODUCTION

Contrastive language-image pretraining (CLIP) (Radford et al., 2021) has become a widely adopted vision–language model, achieving strong zero-shot recognition by comparing image features with text prompts in a shared embedding space. Its scalability (Jia et al., 2021) and adaptability through prompting or ensembling (Zhou et al., 2022; Wortsman et al., 2022) have established it as a foundation model for open-world scenarios where labeled data are scarce. Although CLIP demonstrates impressive generalization ability, it is highly vulnerable to adversarial attacks: tiny pixel-level perturbations, often imperceptible to humans, can cause confident misclassifications and severe drops in performance (Goodfellow et al., 2014; Kurakin et al., 2018; Madry et al., 2017). This contrast between strong zero-shot generalization and fragile robustness motivates the study of adversarial reliability in vision–language models.

Recent efforts on *zero-shot adversarial robustness* aim to enhance CLIP's resistance to adversarial perturbations while preserving zero-shot generalization (Mao et al., 2022; Schlarmann et al., 2024; Xing et al., 2025; Zhang et al., 2025). Formally, the task assumes that only the image encoder is adversarially fine-tuned, while the text encoder remains fixed and provides stable semantic anchors. Existing methods fine-tune the attacked encoder on labeled data to balance clean accuracy and adversarial robustness, and then evaluate transferability to unseen zero-shot datasets (Yu et al., 2024; Wang et al., 2024; Li et al., 2024). A common strategy is to align adversarial features directly to the ground-truth text embedding, which provides strong discriminative supervision but disregards the relative geometry among neighboring classes. As illustrated in Figure 1a (left), the adversarial alignment is enforced only toward the ground-truth text embedding, effectively pulling features along an unconstrained direction and disregarding the relative geometry of neighboring embeddings. However, these relations are essential as they encode inherent data ambiguity, such as semantic overlap between categories or the presence of multiple objects within a single image. Such ambiguity can be naturally interpreted as a form of predictive *uncertainty*. This single-anchor alignment pro-

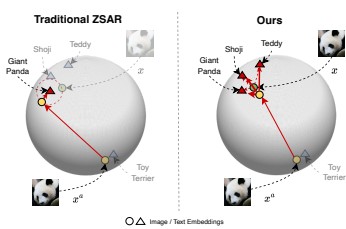
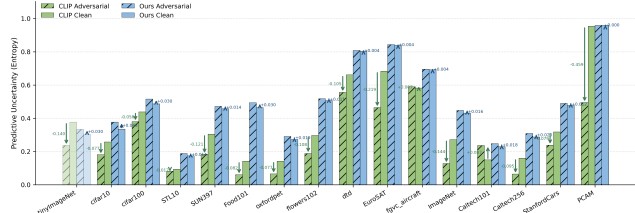

(a) Conceptual illustration.  (b) Predictive Uncertainty under AutoAttack ($\epsilon = 1/255$).

Figure 1: (a) **Conceptual illustration of hypersphere geometry**. Traditional anchor-based zero-shot adversarial robustness (ZSAR) methods align features only to the ground-truth class, while our method preserves inter-class geometry via distributional calibration. (b) **Predictive uncertainty on 16 datasets**. CLIP shows reduced entropy on adversarial inputs, whereas our method UCAT restores calibrated uncertainty. Arrows and numbers show uncertainty change (direction, magnitude).

vides strong discriminative supervision but neglects the underlying uncertainty structure, which can limit generalization under adversarial perturbations.

While previous methods mostly focus on aligning the predicted logits, we argue that they overlook an essential phenomenon, that is, a systematic miscalibration in CLIP's predictive uncertainty under adversarial perturbations. Figure 1b compares entropy-based uncertainty on clean (solid) and adversarial (striped) inputs across multiple datasets. Strikingly, in many cases, the uncertainty of adversarial predictions is lower than that of clean predictions, contradicting the widely held expectation that uncertainty should increase with input difficulty or distributional shift (Guo et al., 2017; Hendrycks & Gimpel, 2016; Ovadia et al., 2019). This anomaly indicates that CLIP not only fails to maintain robustness but also produces spuriously confident predictions when attacked. Such behavior highlights a critical reliability gap beyond accuracy, underscoring the need to calibrate uncertainty in adversarial fine-tuning.

To address both the structural and calibration issues, we propose an Uncertainty-Calibrated Adversarial fine-Tuning (UCAT) framework for CLIP. UCAT operates by regularizing entire Dirichlet distributions rather than anchoring to a single class, thereby preserving inter-class semantic relations while calibrating the overall strength of predictive evidence. This is achieved by reparameterizing CLIP's logits as concentration parameters of a Dirichlet distribution, yielding a unified representation for holistic alignment under perturbations. The quantitative effect of UCAT is shown in Figure 1b: compared to vanilla CLIP, our fine-tuned model achieves calibrated uncertainty levels, restoring a consistent ordering: *original CLIP w/ clean img.* < *fine-tuned CLIP w/ clean img.* < *fine-tuned CLIP w/ adversarial img.*, which faithfully reflects increasing input difficulty. The main contributions of this work can be summarized as follows:

1) **Dirichlet-based formulation of CLIP.** We reformulate CLIP's logits as concentration parameters of a Dirichlet distribution, providing a theoretically justified and closed-form approach to estimate predictive uncertainty.

2) **Uncertainty-Calibrated Adversarial fine-Tuning (UCAT).** We propose a novel uncertainty-calibrated adversarial fine-tuning method that regularizes entire Dirichlet distributions to jointly preserve inter-class relations and calibrate evidence strength.

3) **Extensive empirical validation.** Across 16 single-label benchmarks and the multi-label dataset MS-COCO, we show that our method effectively calibrates uncertainty under attack while maintaining strong clean accuracy and competitive adversarial robustness.

## 2 RELATED WORK

**Zero-shot Adversarial Robustness.** A series of works have advanced zero-shot adversarial robustness (ZSAR) for CLIP by adapting adversarial fine-tuning to the vision–language setting. TeCoA (Mao et al., 2022) pioneered text-guided adversarial fine-tuning with a contrastive loss, aligning adversarial features to ground-truth text prototypes. FARE (Schlarmann et al., 2024) argued that restricting alignment to a single label undermines zero-shot generalization and instead

enforced feature consistency between clean and adversarial representations. Subsequent methods further extended this line with prediction-level (Wang et al., 2024) or attention-level (Yu et al., 2024) regularization. However, all of these approaches adopt the *single-anchor strategy*, which inevitably drives training along an unconstrained direction and disregards the relative geometry of neighboring embeddings. In contrast, we reformulate CLIP logits as Dirichlet evidence, allowing uncertainty to be explicitly calibrated while preserving both semantic structure and confidence strength. This leads to stronger adversarial robustness and improved transfer in open-world settings.

**Uncertainty Calibration.** Uncertainty estimation has been widely explored in settings such as out-of-distribution detection (Hendrycks & Gimpel, 2016; Ovadia et al., 2019), adversarial training (Malinin & Gales, 2019), and large language models (Kuhn et al., 2023). A central challenge is calibration: ideally, uncertainty should increase under harder inputs or distributional shift, yet empirical studies have shown that adversarial predictions often appear spuriously confident (Guo et al., 2017; Hendrycks & Gimpel, 2016; Ovadia et al., 2019). Dirichlet Prior Networks (Malinin & Gales, 2018; Ulmer et al., 2021) addressed this by regularizing logits into Dirichlet parameters, enforcing higher uncertainty on adversarial (Sensoy et al., 2020; Malinin & Gales, 2019) or out-of-distribution samples (Yoon & Kim, 2024). However, in such models the absolute magnitude of evidence is largely an artifact of training and lacks intrinsic meaning. In contrast, CLIP's large-scale contrastive pre-training endows its logits with semantically meaningful absolute strength, which we exploit by reformulating them as Dirichlet evidence. This yields a natural decomposition of predictive uncertainty into *aleatoric uncertainty* (AU), reflecting ambiguity across semantically related classes, and *epistemic uncertainty* (EU), reflecting limited evidence or distributional shift (Ulmer et al., 2021; Ma et al., 2025). To the best of our knowledge, no prior work has established such a theoretical account of uncertainty in CLIP. We fill this gap by proving the Dirichlet structure of CLIP logits and leveraging it for uncertainty-calibrated adversarial robustness.

## 3 PRELIMINARY

### 3.1 CONTRASTIVE LEARNING OBJECTIVE AND ZERO-SHOT CLASSIFICATION

**Contrastive Learning Objective.** Contrastive learning underlies large-scale vision–language models such as CLIP (Radford et al., 2021). Let $f_\theta : \mathcal{X}_{\text{img}} \to \mathbb{R}^d$, $g_\phi : \mathcal{X}_{\text{txt}} \to \mathbb{R}^d$ denote the image and text encoders, where $d$ is the dimension of the embedding space. For an image–text pair $(x_i^{\text{img}}, x_i^{\text{txt}})$, the embeddings are normalized onto the unit hypersphere $\mathbb{S}^{d-1}$: $v_i = f_\theta(x_i^{\text{img}})/\|f_\theta(x_i^{\text{img}})\|_2$, $t_i = g_\phi(x_i^{\text{txt}})/\|g_\phi(x_i^{\text{txt}})\|_2$. The similarity between image $i$ and text $j$ can be expressed in two directional forms: $\ell_{ij}^{v \to t} = \langle v_i, t_j \rangle / \tau$, $\ell_{ij}^{t \to v} = \langle t_i, v_j \rangle / \tau$, where $\tau > 0$ is a learnable temperature parameter. Given a batch of $N$ aligned pairs, the symmetric InfoNCE objective is

$$\mathcal{L}_{\text{CLIP}} = -\frac{1}{2N} \sum_{i=1}^{N} \left[ \log \frac{\exp(\ell_{ii}^{v \to t})}{\sum_{j=1}^{N} \exp(\ell_{ij}^{v \to t})} + \log \frac{\exp(\ell_{ii}^{t \to v})}{\sum_{j=1}^{N} \exp(\ell_{ij}^{t \to v})} \right]. \tag{1}$$

**Zero-shot Classification.** Benefiting from its self-supervised contrastive learning objective, CLIP exhibits strong zero-shot transfer capability for open-vocabulary recognition (Jia et al., 2021; Yao et al., 2021; Zhai et al., 2022; Zhou et al., 2022). At inference, classification is formulated as retrieving the most relevant text prompt for a given image, where only the image-to-text similarity $\ell^{v \to t}$ is evaluated. Each class label $c_k$ ($k = 1, \ldots, C$, where $C$ is the number of candidate classes) is converted into a natural-language prompt (e.g., "This is a photo of a dog"), which is encoded and normalized to yield a class prototype $t_k \in \mathbb{S}^{d-1}$. For a test image $x$, the normalized embedding is $v(x) = f_\theta(x)/\|f_\theta(x)\|_2$, and the logit for class $c_k$ is $\ell_k^{v \to t}(x) = \langle v(x), t_k \rangle / \tau$. The predictive distribution over classes is obtained via the softmax

$$p^{\text{CLIP}}(y = k \mid x) = \frac{\exp(\ell_k^{v \to t}(x))}{\sum_{j=1}^{C} \exp(\ell_j^{v \to t}(x))}. \tag{2}$$

This formulation enables recognition of categories unseen during training, relying solely on the shared image–text embedding space.

## 3.2 Adversarial Attacks.

Adversarial attacks perturb inputs with small, often imperceptible changes to mislead a model. Given an image $x$ with label $y$, an adversarial example is constructed as $x^a = x + \delta, \|\delta\|_q \leq \epsilon$, where $\epsilon$ bounds the perturbation magnitude under $\ell_q$-norm. A canonical method is *Projected Gradient Descent* (PGD, Madry et al., 2017), which iteratively updates

$$x_{t+1}^a = \Pi_{B_\epsilon(x)}\Big(x_t^a + \alpha \operatorname{sign}\big(\nabla_x \mathcal{L}(F_\varphi(x_t^a), y)\big)\Big), \tag{3}$$

where $t$ is the iteration index, $\alpha$ is the step size, $F_\varphi$ is the target model, and $\Pi_{B_\epsilon(x)}$ projects the perturbed point back into the $\epsilon$-ball around $x$. Intuitively, PGD moves the input a small step in the direction that most increases the loss, then clips it to stay within the allowed perturbation range, repeating this process until the attack succeeds.

## 3.3 Uncertainty Estimation via Evidence

**Dirichlet Parameterization with Evidence.** In evidential deep learning (EDL), predictive uncertainty is modeled explicitly by placing a *Dirichlet distribution* over class probabilities rather than predicting a single categorical distribution (Sensoy et al., 2018; Malinin & Gales, 2018; Ulmer et al., 2021). For a $C$-class problem, the network outputs non-negative concentration parameters $\alpha = (\alpha_1, \ldots, \alpha_C) \in \mathbb{R}_+^C$, typically expressed as $\alpha_k = e_k + 1, e_k \geq 0$, where $e_k$ denotes the evidence assigned to class $k$. In the original EDL formulation, this ensures $\alpha_k \geq 1$ so that zero evidence corresponds to a uniform prior. The induced Dirichlet distribution is

$$\operatorname{Dir}(\pi; \alpha) = \frac{1}{B(\alpha)} \prod_{k=1}^C \pi_k^{\alpha_k - 1}, \quad B(\alpha) = \frac{\prod_{k=1}^C \Gamma(\alpha_k)}{\Gamma(\alpha_0)}, \quad \alpha_0 = \sum_{k=1}^C \alpha_k, \tag{4}$$

where $\pi = (\pi_1, \ldots, \pi_C)$ is a probability on the $(C-1)$-simplex and $B(\alpha)$ is the polynomial Beta function. Importantly, $\alpha_0$ quantifies the total evidence and serves as the precision of the distribution.

The non-negativity of $\alpha$ is typically enforced by activation functions such as ReLU, Softplus, or exponential mapping used in prior works (Yoon & Kim, 2024; Malinin & Gales, 2019). In particular, under the exponential parameterization with unconstrained logits $z(x) \in \mathbb{R}^C$ and $\alpha_k(x) = \exp(z_k(x))$, the predictive categorical distribution is obtained as the expectation under the Dirichlet:

$$p(y = k \mid x) := \mathbb{E}_{\pi \sim \operatorname{Dir}(\alpha(x))}[\pi_k] = \frac{\alpha_k(x)}{\alpha_0(x)} \overset{\alpha_k = \exp(z_k)}{=} \frac{\exp(z_k(x))}{\sum_{j=1}^C \exp(z_j(x))}. \tag{5}$$

**Closed-Form Uncertainty Decomposition.** The Dirichlet parameterization not only provides a probability distribution but also admits a closed-form decomposition of predictive uncertainty into two complementary components, aleatoric and epistemic (Der Kiureghian & Ditlevsen, 2009; Kendall & Gal, 2017; Hüllermeier & Waegeman, 2021).

*Aleatoric uncertainty (AU)* captures ambiguity inherent in the data. In vision–language models, this may arise from factors such as semantic overlap between classes (e.g., "wolf" vs. "dog") or noisy image–text pairs where multiple labels are plausible (Ulmer et al., 2021; Ma et al., 2025; Ji et al., 2023). Formally, AU reflects how probability mass is distributed across classes and is quantified by the expected Shannon entropy of the categorical distribution under the Dirichlet:

$$\operatorname{AU}(x) = \mathbb{E}_{\pi \sim \operatorname{Dir}(\alpha)}\big[H(\pi)\big] = -\sum_{k=1}^C \frac{\alpha_k}{\alpha_0}\Big(\psi(\alpha_k + 1) - \psi(\alpha_0 + 1)\Big), \tag{6}$$

where $\psi(\cdot)$ denotes the digamma function.

*Epistemic uncertainty (EU)* arises from limited evidence or distributional shift (Hendrycks & Gimpel, 2016; Sensoy et al., 2018). It reflects the overall reliability of the prediction: when the total evidence $\alpha_0$ is small, the model should be considered untrustworthy. Following prior work (Charpentier et al., 2020; Ulmer et al., 2021; Ma et al., 2025), a widely adopted closed-form proxy is

$$\operatorname{EU}(x) = \frac{C}{\alpha_0 + C}, \tag{7}$$

which increases as $\alpha_0$ decreases.

In summary, AU reflects ambiguity in the predictive distribution across classes, while EU captures uncertainty from insufficient evidence or distributional shift. Both can be computed directly from the Dirichlet parameters, enabling efficient uncertainty estimation in a single forward pass.

# 4 DIRICHLET REFORMULATION OF CLIP

Comparing CLIP's zero-shot probability in Equation 2 with the Dirichlet expectation in Equation 5 reveals a structural correspondence: both are softmax operations over a set of logits. This motivates a *non-trivial* identification that reinterprets CLIP logits as *evidence* governing a Dirichlet distribution (Definition 4.1). This identification is non-trivial for three reasons: (i) it satisfies the validity of Dirichlet evidence with tight bounds and strict monotonicity (Lemma 4.2); (ii) it exactly recovers CLIP's predictive rule exactly under a specific calibration (Lemma 4.3); and (iii) preserves logit order while exposing a tunable temperature for calibration (Corollary 4.3.1).

**Definition 4.1** (Concentration Parameter). *Let $v(x), t_k \in \mathbb{S}^{d-1}$ be unit-normalized image/text embeddings and $\ell_k^{v \to t}(x) = \langle v(x), t_k \rangle / \tau$ the CLIP logit with temperature $\tau > 0$. We define Dirichlet concentration parameters by*

$$\alpha_k(x) = \exp\big(h(\ell_k^{v \to t}(x))\big), \qquad h(\ell) = \frac{\tau\,\ell + 1}{\tau'}, \tag{8}$$

*where $\tau' > 0$ is a calibration coefficient.*

**Remark** (Construction rationale). Since $\tau\,\ell_k^{v \to t}(x) = \langle v(x), t_k \rangle \in [-1, 1]$, we shift the cosine similarity by $+1$ so that its range becomes $[0, 2]$. A calibration coefficient $\tau' > 0$ is introduced to rescale. Applying the exponential guarantees positivity while preserving logit order and remaining compatible with softmax geometry.

**Lemma 4.2** (Validity of Dirichlet Evidence). *Under Definition 4.1, for all $k$:*

1. *$\alpha_k(x) \geq 1$ and $\alpha_k(x) \in [1, \exp(2/\tau')]$;*

2. *$\alpha = \exp(h(\ell))$ is strictly increasing.*

**Remark** ($\alpha_k \geq 1$ in EDL). As introduced in Section 3.3, the classical EDL formulation enforces $\alpha_k \geq 1$ by parameterizing $\alpha_k = e_k + 1$ with non-negative evidence (Sensoy et al., 2018; 2020). We adopt the same restriction for two reasons: (i) digamma- and trigamma-based uncertainty measures become unstable as $\alpha_k$ approaches 0 (Minka, 2000), and (ii) Dirichlet distributions with $\alpha_k < 1$ produce corner-seeking samples (Telgarsky, 2013), concentrating on a few classes even under weak evidence. This violates the common principle that uncertainty should grow as inputs become harder or deviate from the training distribution. Accordingly, our reformulation guarantees $\alpha_k \geq 1$; all subsequent analysis and experiments are under this regime. Proof is provided in Appendix D.1.

**Lemma 4.3** (Exact Equivalence at $\tau = \tau'$). *Let $s = \tau / \tau'$. If $s = 1$ (equivalently $\tau' = \tau$), the Dirichlet expectation equals to CLIP's softmax:*

$$p_k^{\mathrm{Dir}}(x) = \frac{\alpha_k}{\sum_j \alpha_j} = \frac{\exp(h(\ell_k))}{\sum_j \exp(h(\ell_j))} = \mathrm{softmax}\big(\ell(x)\big)_k = p_k^{\mathrm{CLIP}}(x). \tag{9}$$

**Remark** (Significance of exact equivalence). Lemma 4.3 shows that when $\tau' = \tau$, the Dirichlet expectation coincides exactly with CLIP's softmax prediction. This equivalence is not incidental: it demonstrates that CLIP's original training loss in Equation 1 implicitly optimizes a Dirichlet-based model of evidence. Hence, our reformulation is not an ad hoc construction but a faithful probabilistic interpretation of CLIP's logits. A complete proof is provided in Appendix D.2.

**Corollary 4.3.1** (General form and invariances). *For arbitrary $\tau' > 0, s = \tau / \tau' > 0, p^{\mathrm{Dir}}(x) = \mathrm{softmax}\big(s\,\ell(x)\big)$. Hence*

$$\arg\max_k p_k^{\mathrm{Dir}}(x) = \arg\max_k p_k^{\mathrm{CLIP}}(x), \tag{10}$$

*while the entropy of the distribution can be smoothly tuned by $s$: larger $s$ yields sharper predictions, smaller $s$ yields flatter ones.*

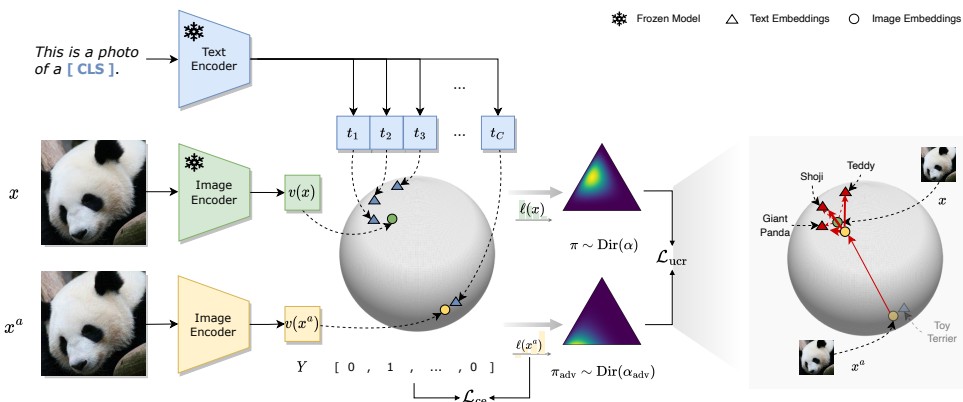

Figure 2: **Overview of our uncertainty calibration adversarial fine-tuning framework.** Clean and adversarial images are encoded by CLIP's image encoder, while text prompts are processed by the frozen text encoder. Our training objective combines the text-guided contrastive loss with an uncertainty calibration regularization term that aligns adversarial Dirichlets with the original clean distributions, thereby preserving semantic relations and calibrating evidence strength.

**Remark** (Connection to uniformity–tolerance in contrastive learning). In contrastive learning, the temperature regulates the separation strength among negatives. A *smaller* softmax temperature (larger $s$) encourages *uniformity* on the hypersphere by enforcing stronger separation, while a *larger* temperature (smaller $s$) increases *tolerance* to near-semantic neighbors (Wang & Isola, 2020; Wang & Liu, 2021). We set $\tau' = 0.07$, yielding $s < 1$ and thus softer predictions that increase tolerance to semantically related negatives. Our choice also matches the canonical temperature used in MoCo (He et al., 2020) and other contrastive frameworks (Radford et al., 2021; Jia et al., 2021), which has been widely validated for stable and well-separated representations, with sensitivity analysis in Appendix F.3. Such a setting preserves CLIP's intrinsic semantic structure and is particularly beneficial for adversarial fine-tuning, where calibrated tolerance improves zero-shot robustness without harming the model's original generalization ability. Proof is deferred to Appendix D.2.1.

This reformulation establishes a principled mapping from CLIP logits to Dirichlet evidence, serving several important implications. First, it naturally admits provides *closed-form uncertainty decomposition* (Section 3.3), enabling direct and decoupled quantification of aleatoric and epistemic components without auxiliary sampling. Second, it offers *principled calibration*, since the calibration coefficient adjusts confidence sharpness without altering prediction accuracy, allowing a controllable trade-off between uniformity and tolerance. Finally, it ensures *semantic fidelity*. The reformulation not only recovers CLIP's predictive rule in the exact equivalence case but also supports optimization over a Dirichlet distribution that preserves relative geometry and absolute evidence strength. These properties lay the foundation for the adversarial fine-tuning objectives introduced in the next section.

## 5 UNCERTAINTY CALIBRATION ADVERSARIAL FINE-TUNING OBJECTIVE

To mitigate the misaligned semantics and unreliable confidence introduced by adversarial perturbations, we propose an *Uncertainty Calibration Adversarial fine-Tuning (UCAT)* objective. The key insight builds on our reformulation: mapping CLIP logits to Dirichlet evidence yields closed-form uncertainty decomposition with principled calibration, while retaining fidelity to the semantic geometry of the embedding space. UCAT exploits this property by aligning the Dirichlet distributions of adversarial and clean samples, correcting distributional shift while simultaneously preserving *semantic relations* and *calibrated confidence*.

As illustrated in Figure 2, our method adopts a CLIP-based adversarial fine-tuning pipeline with a frozen text encoder and a trainable image encoder. Clean samples $x$ and their adversarial counterparts $x^a$ (generated via $\ell_\infty$-PGD (Madry et al., 2017)) are encoded into the joint embedding space, and their logits are reformulated as Dirichlet parameters, denoted $\alpha$ and $\alpha_{adv}$. The clean distribution $Dir(\alpha)$ captures the generalized semantics from pre-training, whereas $Dir(\alpha_{adv})$ may shift toward distorted or overconfident states. To correct this mismatch, we introduce an *uncertainty calibration*

Table 1: **Zero-shot adversarial robustness on multi-label dataset MS-COCO (Lin et al., 2014).** All models are adversarially trained on TinyImageNet with the FARE (Schlarmann et al., 2024) 10-step PGD ($\epsilon = 2/255$) setting and evaluated under CW-100 (Carlini & Wagner, 2017) attacks. We report micro-averaged Precision (P), Recall (R), and F1-score (F1) at top-3 and top-5 predictions, together with mean Average Precision (mAP), under adversarial conditions. Best and second-best are in **bold** and underline.

| Methods | P@3 | R@3 | F1@3 | P@5 | R@5 | F1@5 | mAP |
|---|---|---|---|---|---|---|---|
| CLIP (Radford et al., 2021) | 17.72 | 25.21 | 25.85 | 25.52 | 20.15 | 34.44 | 25.42 |
| TeCoA (Mao et al., 2022) | 30.23 | 30.99 | 30.60 | 22.67 | 38.73 | 28.59 | 37.32 |
| FARE (Schlarmann et al., 2024) | 33.45 | 34.30 | 33.86 | 26.04 | 44.49 | 32.84 | 29.18 |
| PMG-AFT (Wang et al., 2024) | 32.32 | 33.15 | 32.72 | 25.40 | 43.40 | 32.04 | 29.75 |
| TGA-ZSR (Yu et al., 2024) | 32.95 | 33.79 | 33.36 | 24.58 | 41.99 | 31.00 | **38.23** |
| UCAT (Ours) | **36.58** | **37.52** | **37.04** | **28.27** | **48.32** | **35.67** | 37.60 |

*regularization* objective, defined as the KL divergence between the two distributions:

$$\mathcal{L}_{\text{ucr}} = \text{KL}(\text{Dir}(\alpha_{\text{adv}}) \,\|\, \text{Dir}(\alpha)) . \tag{11}$$

Since both AU and EU are closed-form functions of Dirichlet parameters (Sec. 3.3), minimizing $\mathcal{L}_{\text{ucr}}$ aligns adversarial predictions with their clean counterparts in terms of *inter-class relations* (AU) and *evidence magnitude* (EU), thereby preventing collapse into spuriously confident errors. Complementarily, the text-guided cross-entropy loss

$$\mathcal{L}_{\text{ce}} = -\log \frac{\exp\left(\langle v(x^a), t_y \rangle / \tau\right)}{\sum_{j=1}^{C} \exp\left(\langle v(x^a), t_j \rangle / \tau\right)}, \tag{12}$$

anchors adversarial embeddings to the ground-truth prototype $t_y$, providing discriminative supervision that stabilizes training and improves accuracy. The final objective combines both components:

$$\mathcal{L} = \mathcal{L}_{\text{ce}} + \lambda \, \mathcal{L}_{\text{ucr}}, \tag{13}$$

where $\lambda$ balances discriminative alignment and uncertainty calibration. This joint objective combines discriminative supervision via the cross-entropy loss with calibrated uncertainty through distributional alignment, leading to stronger zero-shot adversarial robustness.

## 6 EXPERIMENTS

**Implementational Details and Datasets.** We adopt CLIP-B/32 (Radford et al., 2021) as the backbone and follow TeCoA's training protocol (Mao et al., 2022), comparing zero-shot adversarial robustness against five baselines: CLIP (Radford et al., 2021), TeCoA, FARE (Schlarmann et al., 2024), PMG-AFT (Wang et al., 2024), and TGA-ZSR (Yu et al., 2024). Training and evaluation are conducted under $\ell_\infty$ PGD regimes, including a light setting (2-step, $\epsilon = 1/255$) following (Mao et al., 2022) and a stronger setting (10-step, $\epsilon = 2/255$) following (Schlarmann et al., 2024). Robustness is further assessed using 100-step PGD (Madry et al., 2017), CW (Carlini & Wagner, 2017), and AutoAttack (Croce & Hein, 2020). We set $\lambda = 10^5/\beta$ with $\beta = 2/e^{\tau'}$, and fix $\tau' = 0.07$ following standard contrastive learning practices (Wu et al., 2018; He et al., 2020; Radford et al., 2021; Yeh et al., 2022). Full implementation details and datasets are provided in the Appendix C.

### 6.1 EFFICIENCY ON MULTI-LABEL DATA AMBIGUITY

To assess robustness under data ambiguity, we perform zero-shot evaluation on the multi-label MS-COCO (Lin et al., 2014) dataset (Table 1). All models are fine-tuned on single-label TinyImageNet using PGD, and tested directly on COCO under CW attacks to perturb multiple labels simultaneously. Our method achieves the best top-$k$ precision, recall, and F1, indicating stronger ability to recognize multiple objects within a single image. Compared with label-guided approaches that explicitly align adversarial features to the ground-truth class, both our method and FARE benefit from preserving the intrinsic generalization encoded in CLIP's original features. By further incorporating uncertainty calibration, our method balances semantic fidelity with calibrated confidence, leading to consistently stronger robustness under multi-label ambiguity. While our mAP is also competitive, this metric is easily influenced by low-probability noise from irrelevant categories, making top-$k$ evaluation a more faithful measure of robustness to label ambiguity.

## 6.2 Cross-dataset Evaluation of Zero-shot Adversarial Robustness

Table 2: **Zero-shot adversarial robustness across 16 single-label datasets.** All methods are fine-tuned on TinyImageNet following TGA-ZSR (Yu et al., 2024), adversarial training uses 2-step PGD (Madry et al., 2017) with $\epsilon = 1/255$. *Average* is the mean across datasets. $H$ is the harmonic mean between Clean and the corresponding robust score. Best and second-best are in **bold** and underline.

| | Methods | TinyImageNet | Cifar10 | Cifar100 | STL10 | SUN397 | Food101 | Oxfordpets | Flowers102 | DTD | EuroSAT | FGVC Aircraft | ImageNet | Caltech101 | Caltech256 | StanfordCars | PCAM | Average | H |
|---|---|---|---|---|---|---|---|---|---|---|---|---|---|---|---|---|---|---|---|
| Clean | CLIP (Radford et al., 2021) | 57.96 | 88.03 | 60.45 | 97.03 | 57.26 | 83.89 | 87.41 | 65.49 | 40.64 | 42.66 | 20.16 | 59.15 | 85.32 | 81.73 | 52.02 | 52.08 | 64.45 | |
| | TeCoA (Mao et al., 2022) | 71.24 | 67.56 | 38.26 | 85.89 | 36.01 | 28.23 | 61.30 | 32.04 | 24.95 | 16.13 | 5.19 | 32.89 | 72.16 | 59.00 | 20.28 | 50.11 | 43.83 | |
| | FARE (Schlarmann et al., 2024) | 41.86 | 79.81 | 48.27 | 94.24 | 46.15 | 58.90 | 80.98 | 47.63 | 23.09 | 24.19 | 15.63 | 42.93 | 78.22 | 72.05 | 43.96 | 50.02 | 53.00 | |
| | PMG-AFT (Wang et al., 2024) | 48.60 | 74.73 | 43.59 | 90.41 | 51.70 | 56.52 | 79.40 | 48.43 | 32.45 | 21.76 | 11.79 | 46.74 | 82.49 | 73.59 | 41.21 | 56.13 | 53.72 | |
| | TGA-ZSR (Yu et al., 2024) | 76.60 | 79.18 | 47.37 | 90.65 | 43.10 | 38.90 | 68.44 | 39.81 | 25.69 | 19.70 | 8.82 | 39.27 | 76.42 | 66.31 | 28.44 | 49.92 | 49.91 | |
| | UCAT (Ours) | 74.46 | 81.81 | 54.45 | 91.88 | 41.06 | 53.58 | 74.16 | 47.57 | 31.92 | 19.29 | 10.95 | 43.20 | 82.39 | 71.53 | 37.32 | 51.20 | 54.17 | |
| PGD | CLIP (Radford et al., 2021) | 0.19 | 9.57 | 3.07 | 23.64 | 0.62 | 0.34 | 0.64 | 1.62 | 2.22 | 0.00 | 0.00 | 0.48 | 5.65 | 7.19 | 0.02 | 0.06 | 3.46 | 6.56 |
| | TeCoA (Mao et al., 2022) | 50.96 | 39.33 | 21.64 | 69.78 | 20.07 | 13.50 | 37.80 | 19.17 | 18.30 | 11.88 | 2.16 | 18.47 | 56.00 | 42.38 | 9.33 | 46.92 | 29.86 | 35.52 |
| | FARE (Schlarmann et al., 2024) | 3.78 | 7.83 | 2.80 | 48.18 | 5.66 | 2.45 | 10.93 | 6.52 | 5.75 | 0.08 | 0.54 | 5.20 | 33.21 | 20.70 | 2.31 | 48.97 | 12.81 | 20.63 |
| | PMG-AFT (Wang et al., 2024) | 19.18 | 51.39 | 27.23 | 72.63 | 20.05 | 16.88 | 44.59 | 26.43 | 20.05 | 11.49 | 3.21 | 18.09 | 61.13 | 43.46 | 14.80 | 55.52 | 31.63 | 39.82 |
| | TGA-ZSR (Yu et al., 2024) | 50.68 | 42.16 | 22.82 | 72.18 | 21.57 | 16.53 | 39.96 | 22.44 | 17.82 | 11.75 | 2.88 | 20.39 | 65.32 | 50.47 | 15.30 | 48.05 | 31.55 | 38.66 |
| | UCAT (Ours) | 47.56 | 43.81 | 25.16 | 73.83 | 20.44 | 22.86 | 45.11 | 26.79 | 19.47 | 2.99 | 3.45 | 22.22 | 65.32 | 50.47 | 15.30 | 30.37 | 32.20 | 40.39 |
| CW | CLIP (Radford et al., 2021) | 0.14 | 9.91 | 3.34 | 26.01 | 1.16 | 0.51 | 0.87 | 2.03 | 2.55 | 0.01 | 0.00 | 1.10 | 6.82 | 8.17 | 2.32 | 0.04 | 4.06 | 7.64 |
| | TeCoA (Mao et al., 2022) | 50.16 | 38.62 | 20.76 | 69.55 | 18.84 | 12.46 | 37.37 | 18.12 | 17.23 | 11.63 | 2.10 | 17.70 | 55.62 | 41.70 | 9.23 | 46.88 | 29.25 | 35.08 |
| | FARE (Schlarmann et al., 2024) | 4.10 | 4.12 | 2.96 | 43.35 | 6.07 | 3.17 | 15.15 | 5.66 | 4.52 | 0.12 | 1.11 | 5.34 | 32.50 | 20.85 | 4.38 | 48.86 | 12.64 | 20.41 |
| | PMG-AFT (Wang et al., 2024) | 13.16 | 42.10 | 21.31 | 65.69 | 13.12 | 11.43 | 28.05 | 17.53 | 12.55 | 8.51 | 0.99 | 11.72 | 52.84 | 35.68 | 7.06 | 14.26 | 22.25 | 31.47 |
| | TGA-ZSR (Yu et al., 2024) | 50.80 | 42.24 | 22.64 | 71.99 | 20.83 | 16.03 | 40.20 | 21.52 | 16.97 | 11.56 | 2.85 | 20.01 | 57.72 | 45.84 | 11.23 | 48.03 | 31.28 | 38.46 |
| | UCAT (Ours) | 47.08 | 43.30 | 23.92 | 73.55 | 19.20 | 21.68 | 45.38 | 24.95 | 17.87 | 2.41 | 3.21 | 21.14 | 64.63 | 49.54 | 14.75 | 29.89 | 31.41 | 39.76 |
| Auto Attack | CLIP (Radford et al., 2021) | 0.00 | 2.54 | 1.11 | 3.18 | 0.04 | 0.03 | 0.02 | 0.17 | 0.23 | 0.04 | 0.10 | 0.26 | 0.07 | 0.12 | | | 0.51 | 1.01 |
| | TeCoA (Mao et al., 2022) | 49.44 | 37.87 | 20.45 | 69.31 | 17.41 | 12.19 | 36.58 | 17.81 | 17.29 | 11.42 | 1.86 | 17.19 | 54.95 | 41.19 | 8.16 | 46.79 | 28.74 | 34.72 |
| | FARE (Schlarmann et al., 2024) | 0.12 | 0.03 | 0.21 | 10.18 | 0.84 | 0.19 | 0.93 | 0.60 | 1.92 | 0.07 | 0.06 | 0.86 | 10.26 | 5.59 | 0.21 | 5.15 | 2.33 | 4.45 |
| | PMG-AFT (Wang et al., 2024) | 8.22 | 41.86 | 21.18 | 65.45 | 7.95 | 7.34 | 18.94 | 12.59 | 3.13 | 7.17 | 0.51 | 7.90 | 44.91 | 28.29 | 3.22 | 7.41 | 17.88 | 26.83 |
| | TGA-ZSR (Yu et al., 2024) | 49.26 | 40.92 | 21.75 | 71.55 | 19.88 | 15.32 | 38.84 | 20.98 | 17.02 | 11.26 | 2.34 | 19.12 | 57.11 | 45.16 | 9.87 | 48.00 | 30.52 | 37.88 |
| | UCAT (Ours) | 45.80 | 42.32 | 23.03 | 73.15 | 18.26 | 20.52 | 44.02 | 24.54 | 18.14 | 2.26 | 2.61 | 20.15 | 63.73 | 48.66 | 12.60 | 29.51 | 30.58 | 39.09 |

To verify the effectiveness of our approach under single-label settings, we analyze results across 16 datasets (Table 2). Our method achieves consistently strong performance, ranking best or second-best in nearly all cases. When trained with a single PGD regime, it generalizes effectively to multiple adversarial attacks while maintaining both the highest clean accuracy and adversarial robustness. The only exceptions are two domain-specific datasets (PCAM (Veeling et al., 2018) and EuroSAT (Helber et al., 2019)), which exhibit the highest predictive uncertainty (high PU in Fig. 1b, high AU in Fig. 5, and low EU in Fig. 6) and strong semantic overlap. These characteristics reflect a substantial departure from CLIP's natural-image pre-training domain, resulting in inherently weaker clean semantic geometry. Consequently, UCAT has less reliable structure to preserve through Dirichlet alignment, naturally limiting the magnitude of improvement. Nevertheless, UCAT remains stable on these domain-shifted datasets and achieves state-of-the-art robustness on the majority of natural-image benchmarks, where CLIP provides strong clean semantic structure and our Dirichlet alignment is most effective. We further extend our evaluation to larger-scale training, stronger attack settings, and additional ablations, with results reported in Appendix F.

## 6.3 Ablation and Parameter Sensitivity

We conduct an ablation study to disentangle the role of different loss components in Table 3. Using only the text-guided cross-entropy $\mathcal{L}_{ce}$ already improves robustness compared to vanilla CLIP by providing discriminative supervision. Aligning probability distributions at the softmax level further improves performance by preserving relative class geometry, but this approach discards absolute evidence magnitude due to normalization, limiting its effect. In contrast, our Dirichlet-level alignment preserves both relative relationships and absolute evidence strength, thereby calibrating uncertainty more effectively. This joint design yields the best balance between clean accuracy and adversarial robustness across diverse datasets and attacks.

Table 3: **Ablation study.** Trained on TinyImageNet with 1-step PGD and evaluated under 100-step PGD, CW, and AutoAttack (AA) with $\epsilon = 1/255$. Results are averaged over 16 datasets. Best and second-best are in **bold** and underline.

| Methods | Clean | PGD | CW | AA |
|---|---|---|---|---|
| CLIP | 64.45 | 0.05 | 4.06 | 0.51 |
| $\mathcal{L}_{ce}$ | 43.83 | 29.86 | 29.25 | 28.74 |
| $\mathcal{L}_{ce}$+KL($p(x^a)\|p(x)$) | 45.05 | 29.98 | 29.28 | 28.80 |
| $\mathcal{L}_{ce}$+KL(Dir($\alpha_{adv}\|\alpha$)) | **54.17** | **32.20** | **31.41** | **30.58** |

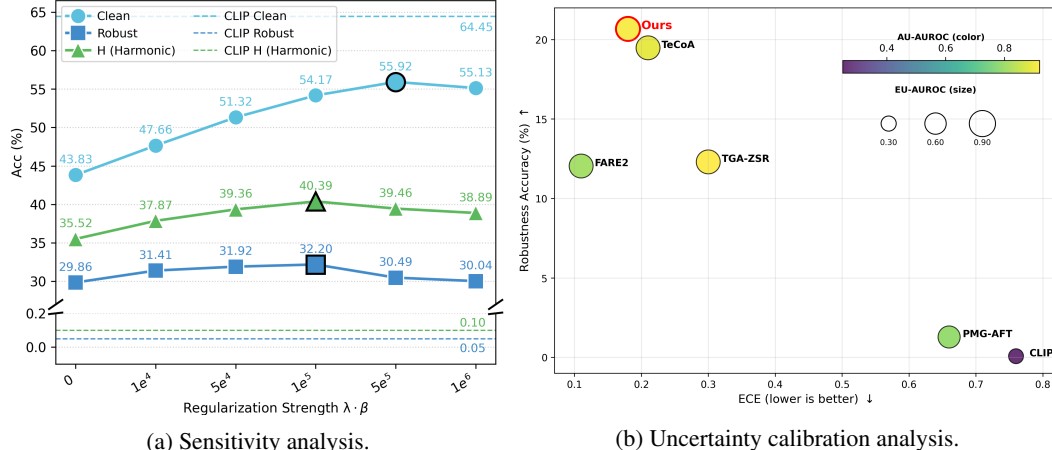

(a) Sensitivity analysis.

(b) Uncertainty calibration analysis.

Figure 3: (a) **Sensitivity analysis of the regularization strength** $\lambda$. We evaluate $\lambda \cdot \beta \in \{10^4, 5 \times 10^4, 10^5, 5 \times 10^5, 10^6\}, \beta = 2/e^{\tau'}$ on all 16 datasets, reporting averages of clean accuracy, PGD-100 (Madry et al., 2017) robustness, and their harmonic mean. (b) **Comprehensive evaluation averaged over 16 datasets under strong adversarial training (PGD-10, $\epsilon = 2/255$) and AutoAttack (Croce & Hein, 2020) testing.** X-axis shows calibration error (ECE, lower is better), while Y-axis shows robustness accuracy. Bubble color indicates AU-AUROC and bubble size indicates EU-AUROC, reflecting the discriminative power of aleatoric and epistemic uncertainty.

Varying $\lambda$ reveals stable performance across a broad range, with the best trade-off at $10^5/\beta$ (Fig. 3a). At this point, clean accuracy reaches 54.17%, robust accuracy reaches 32.20%, and the harmonic mean peaks at 40.39%. This setting is particularly meaningful, as it balances $\mathcal{L}_{ce}$ and $\mathcal{L}_{ucr}$ to contribute comparably during training. Smaller $\lambda$ under-regularizes and limits the benefit of uncertainty calibration, while larger values overweight distributional alignment and degrade robustness.

### 6.4 ROBUSTNESS, CALIBRATION, AND UNCERTAINTY UNDER STRONG ATTACKS

Figure 3b provides a comprehensive evaluation under AutoAttack (Croce & Hein, 2020) with $\epsilon = 2/255$. We report four complementary metrics. *Expected Calibration Error (ECE)* (x-axis) measures how well predicted confidence matches actual correctness (lower is better), while *robustness accuracy* (y-axis) captures the ability to resist adversarial perturbations (higher is better). Bubble color denotes *AU-AUROC*, reflecting how aleatoric uncertainty helps identify errors caused by class ambiguity, and bubble size denotes *EU-AUROC*, reflecting how epistemic uncertainty captures errors due to insufficient evidence. An ideal model should lie toward the top-left of the plot (high robustness, low ECE) with large and bright bubbles (high AU-AUROC and EU-AUROC). Our method is closest to this desirable region: it achieves the highest robustness accuracy, maintains lower calibration error than existing baselines, and exhibits stronger uncertainty discrimination as shown by larger and brighter bubbles. This demonstrates that our uncertainty calibration not only strengthens adversarial robustness but also improves predictive reliability under attack.

## 7 CONCLUSION

In this paper, we identified that adversarial perturbations in zero-shot CLIP not only reduce accuracy but also often suppress predictive uncertainty, leading to severe miscalibration. To address this, we reformulated CLIP logits as Dirichlet concentration parameters, yielding a representation that preserves both semantic structure and confidence strength. Building on this foundation, we introduced an uncertainty calibration adversarial finetuning method that aligns the Dirichlet distributions of clean and perturbed samples, ensuring robustness preservation and calibrated uncertainty. Extensive experiments demonstrate that our approach improves adversarial robustness, handles data ambiguity, and provides reliable uncertainty estimates. Beyond CLIP, our contrastive-theoretic perspective suggests a principled way to analyze and extend uncertainty modeling to other contrastive learning frameworks.

ETHICS STATEMENT

This work uses only computational methods and publicly available datasets, with no human subjects or private data. It follows the ICLR Code of Ethics, with no conflicts of interest. While acknowledging potential dual-use concerns, we stress responsible deployment and adhere to research integrity. All methods and results are reported transparently to support reproducibility.

REPRODUCIBILITY STATEMENT

We provide implementation details in the appendix to support reproduction of the main results

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

## A  LLM USAGE DISCLOSURE

We used large language models (e.g., ChatGPT, GPT-5) solely for language editing and clarity improvement of the manuscript. All research ideas, experimental design, implementation, analyses, and conclusions were fully developed and verified by the authors.

## B  EXTENDED RELATED WORK

### B.1  CONTRASTIVE LEARNING

Self-supervised contrastive learning has proven highly effective in learning transferable representations across tasks such as classification (Chen et al., 2020; Grill et al., 2020), detection (Xie et al., 2021a;b), and segmentation (He et al., 2020; Caron et al., 2021). Building on this foundation, CLIP (Radford et al., 2021) extends contrastive pre-training to large-scale image–text pairs and achieves remarkable zero-shot recognition performance. Its scalability (Jia et al., 2021) and adaptability through fine-tuning or ensembling (Zhou et al., 2022; Wortsman et al., 2022) further establish vision–language models as a powerful paradigm for open-world scenarios where labeled data is scarce.

Recent theoretical analyses further clarify why contrastive objectives are effective. The alignment–uniformity framework (Wang & Isola, 2020) explains how positive pairs encourage semantic consistency while negatives enforce diversity on the hypersphere, and subsequent studies refine our understanding of how loss geometry and temperature schedules shape representation quality (Yeh et al., 2022). Beyond accuracy, contrastive pre-training has also been examined from the perspective of robustness. Prior work shows that robustness may not automatically transfer from contrastive pre-training to downstream fine-tuning (Mao et al., 2022), motivating approaches that explicitly integrate contrastive signals into adversarial training or synthetic data generation (Ouyang et al., 2023). Together, these studies indicate that contrastive learning not only underpins the success of large-scale vision–language models, but also implicitly encodes semantic geometry and confidence cues, laying the foundation for uncertainty-aware robustness.

### B.2  ADVERSARIAL TRAINING AND UNCERTAINTY-BASED ADVERSARIAL DETECTION

Classical adversarial training approaches such as TRADES (Zhang et al., 2019) establish the theoretically principled trade-off between natural and robust errors, while more recent variants like ACAT (Addepalli et al., 2022) and the latest DKL (Cui et al., 2024) further improve adversarial robustness through efficient perturbation generation or decoupled optimization formulation of KL divergence. However, their formulation inherently assumes a closed vocabulary and a parametric classifier head. When directly applied to CLIP, the supervised objectives disrupt the pretrained image–text geometry and degrade zero-shot generalization.

In parallel, adversarial detection and uncertainty-driven robustness methods such as Prior Networks (Malinin & Gales, 2019) and CCAT (Stutz et al., 2020) impose hand-crafted Dirichlet or mixed target distributions for adversarial inputs. These methods are highly effective in closed-set adversarial detection, where the target uncertainty profile is explicitly defined. Yet, their fixed uncertainty targets override the intrinsic evidential semantics encoded in CLIP's normalized contrastive logits, whose magnitudes reflect meaningful evidence strength acquired during large-scale contrastive pretraining. Consequently, such designs fail to preserve fine-grained inter-class relations that are essential for zero-shot retrieval and open-set alignment.

### B.3  ZERO-SHOT ADVERSARIAL ROBUSTNESS

Adversarial robustness has traditionally been studied through supervised adversarial training, with methods such as PGD-based minimax optimization (Madry et al., 2017) and regularized formulations like TRADES (Zhang et al., 2019) offering strong baselines. However, these approaches rely on labeled data and do not directly address the zero-shot setting of vision–language models. Recent works therefore explore adversarial robustness of CLIP without requiring task-specific supervision. TeCoA (Mao et al., 2022) aligns adversarial features with text prototypes to preserve zero-shot transfer, while FARE (Schlarmann et al., 2024) emphasizes maintaining the original visual embedding

geometry. Other strategies such as PMG-AFT (Wang et al., 2024) and TGA-ZSR (Yu et al., 2024) incorporate prompt-based or gradient-aligned objectives to enhance robustness. Despite their differences, these methods share the challenge of balancing robustness with CLIP's inherent semantic structure, highlighting the need for approaches that explicitly model uncertainty and reliability under adversarial perturbations.

### B.4 UNCERTAINTY ESTIMATION WITH EVIDENCE

Uncertainty estimation has been widely explored to improve the reliability of deep neural networks. Classical approaches include Bayesian neural networks (Blundell et al., 2015), Monte Carlo dropout (Gal & Ghahramani, 2016), and deep ensembles (Lakshminarayanan et al., 2017), which approximate predictive distributions through sampling or model averaging. More recent work in evidential learning proposes to represent predictions as parameters of a Dirichlet distribution (Sensoy et al., 2018; Malinin & Gales, 2018), naturally decomposing predictive uncertainty into aleatoric and epistemic components. This evidential perspective has been applied to tasks such as calibration (Ulmer et al., 2021) and out-of-distribution detection (Yoon & Kim, 2024), demonstrating both theoretical interpretability and empirical effectiveness. In adversarial settings, evidential models have shown promise in capturing distributional shifts and mitigating overconfident errors (Malinin & Gales, 2019). Most recently, evidence-based uncertainty estimation has also been extended to large language models, where LogTokU (Ma et al., 2025) treats logits as Dirichlet evidence to decouple aleatoric and epistemic uncertainty, further underscoring the importance of evidence modeling as a principled framework for reliable predictions.

## C IMPLEMENTATION DETAILS

**Dataset.** The same zero-shot evaluation suite as in other ZSAR baselines (e.g., Mao et al. (2022)): ImageNet/tinyImageNet (Deng et al., 2009), CIFAR10/100 (Krizhevsky et al., 2009), STL10 (Coates et al., 2011), Caltech101 (Fei-Fei et al., 2004), Caltech256 (Griffin et al., 2007), OxfordPets (Parkhi et al., 2012), StanfordCars (Krause et al., 2013), Food101 (Bossard et al., 2014), Flowers102 (Nilsback & Zisserman, 2008), FGVC-Aircraft (Maji et al., 2013), SUN397 (Xiao et al., 2010), DTD (Cimpoi et al., 2014), and two domain-specialized sets PCAM (Veeling et al., 2018) and EuroSAT (Helber et al., 2019). To further assess robustness under semantic ambiguity, we additionally include the multi-label dataset MS-COCO (Lin et al., 2014).

We adopt CLIP-B/32 (Radford et al., 2021) as the backbone and follow TeCoA's optimizer and training schedule (Mao et al., 2022), using a batch size of 256 and 10 training epochs unless otherwise stated. We benchmark five methods: CLIP (Radford et al., 2021), TeCoA (Mao et al., 2022), FARE (Schlarmann et al., 2024), PMG-AFT (Wang et al., 2024), and TGA-ZSR (Yu et al., 2024).

*Training Attacks.* We adopt two regimes: (i) a light regime following TeCoA, using $\ell_\infty$ PGD-2 with $\varepsilon = 1/255$ and step size $\alpha = 1/255$; and (ii) a stronger regime following FARE, using $\ell_\infty$ PGD-10 with $\varepsilon = 2/255$ and step size $\alpha = 2/255$.

*Evaluation Attacks.* Robustness is further assessed using $\ell_\infty$ PGD-100 (Madry et al., 2017) (with the same $\varepsilon$ as the training regime and $\alpha = \varepsilon$), CW-100 (Carlini & Wagner, 2017), and AutoAttack (Croce & Hein, 2020) (the rand version ensembling APGD-CE and APGD-DLR).

*Loss Weights.* We set $\lambda = 10^5/\beta$ with $\beta = 2/e^{\tau'}$, where $\tau' = 0.07$ follows standard contrastive learning practices (Wu et al., 2018; He et al., 2020; Radford et al., 2021; Yeh et al., 2022). Here $\beta$ corresponds to the upper bound of the mapping function $h(\ell)$ that converts logits $\ell$ into non-negative evidence. Using this bound guarantees that $\lambda$ remains numerically stable across different temperature values, preventing uncontrolled scaling when $\tau'$ varies.

## D PROOF OF LEMMA

### D.1 LEMMA 1: VALIDITY OF DIRICHLET EVIDENCE

**Lemma D.1** (Validity of Dirichlet Evidence). *Under Definition 4.1, for all $k$:*

1. $\alpha_k(x) \geq 1$ *and* $\alpha_k(x) \in [1, \exp(2/\tau')]$;

2. $\alpha = \exp(h(\ell))$ *is strictly increasing.*

*Proof.* Since $\|v(x)\|_2 = \|t_k\|_2 = 1$, we have $\langle v(x), t_k \rangle \in [-1, 1]$. By the logit definition, $\tau\, \ell_k^{v \to t}(x) = \langle v(x), t_k \rangle \in [-1, 1]$. Therefore,

$$h(\ell_k^{v \to t}(x)) = \frac{\tau\, \ell_k^{v \to t}(x) + 1}{\tau'} \in \left[ 0,\, \frac{2}{\tau'} \right].$$

Exponentiating yields

$$\alpha_k(x) = \exp\big(h(\ell_k^{v \to t}(x))\big) \in \left[ e^0,\, e^{2/\tau'} \right] = \left[ 1,\, \exp(2/\tau') \right],$$

and both endpoints are attainable when $\langle v(x), t_k \rangle = -1$ and $+1$, respectively.

For monotonicity, differentiate $\alpha_k(x)$ with respect to $\ell_k^{v \to t}(x)$:

$$\frac{d\,\alpha_k(x)}{d\,\ell_k^{v \to t}(x)} = \frac{\tau}{\tau'}\, \exp\Big( \frac{\tau\, \ell_k^{v \to t}(x) + 1}{\tau'} \Big) = \frac{\tau}{\tau'}\, \alpha_k(x) > 0,$$

since $\tau > 0$, $\tau' > 0$, and $\alpha_k(x) > 0$. Hence $\alpha_k$ is strictly increasing in $\ell_k^{v \to t}$, which preserves both strict and non-strict order between any pair of logits.

### D.2 LEMMA 2: CONSISTENCY WITH DIRICHLET EXPECTATIONS

**Lemma D.2** (Exact Equivalence at $\tau = \tau'$). *Let* $s = \tau/\tau'$. *If* $s = 1$ *(equivalently $\tau' = \tau$), the Dirichlet expectation equals to CLIP's softmax:*

$$p_k^{\mathrm{Dir}}(x) = \frac{\alpha_k}{\sum_j \alpha_j} = \frac{\exp(h(\ell_k))}{\sum_j \exp(h(\ell_j))} = \mathrm{softmax}\big(\ell(x)\big)_k = p_k^{\mathrm{CLIP}}(x).$$

*Proof.* From the definition of the Dirichlet expectation in Equation 5,

$$p_k^{\mathrm{Dir}}(x) = \mathbb{E}_{\pi \sim \mathrm{Dir}(\alpha(x))}[\pi_k] = \frac{\alpha_k(x)}{\alpha_0(x)}, \quad \alpha_0(x) = \sum_{j=1}^{C} \alpha_j(x).$$

By construction,

$$\alpha_k(x) = \exp(h(\ell_k^{v \to t}(x))), \quad h(\ell_k^{v \to t}(x)) = \frac{\tau \ell_k^{v \to t}(x) + 1}{\tau'} = \frac{1}{\tau'} + \frac{\tau}{\tau'} \ell_k^{v \to t}(x).$$

Let $s = \tau/\tau' > 0$. Then

$$p_k^{\mathrm{Dir}}(x) = \frac{\exp(1/\tau' + s\, \ell_k^{v \to t}(x))}{\sum_{j=1}^{C} \exp(1/\tau' + s\, \ell_j^{v \to t}(x))} = \frac{\exp(s\, \ell_k^{v \to t}(x))}{\sum_{j=1}^{C} \exp(s\, \ell_j^{v \to t}(x))} = \mathrm{softmax}(s\, \ell^{v \to t}(x))_k,$$

since the additive constant $1/\tau'$ cancels out. When $s = 1$ (equivalently, $\tau' = \tau$), this reduces to

$$p_k^{\mathrm{Dir}}(x) = \mathrm{softmax}(\ell^{v \to t}(x))_k,$$

which matches exactly the original CLIP prediction $p_k^{\mathrm{CLIP}}(x)$.

**Corollary D.2.1** (General form and invariances). *For arbitrary* $\tau' > 0, s = \tau/\tau' > 0$, $p^{\mathrm{Dir}}(x) = \mathrm{softmax}\big(s\, \ell(x)\big)$. *Hence*

$$\arg\max_k p_k^{\mathrm{Dir}}(x) = \arg\max_k p_k^{\mathrm{CLIP}}(x)$$

*while the entropy of the distribution can be smoothly tuned by $s$: larger $s$ yields sharper predictions, smaller $s$ yields flatter ones.*

*Proof.* For any logits $\ell \in \mathbb{R}^C$ and scalar $s > 0$,

$$\arg\max_k \ell_k = \arg\max_k s\ell_k.$$

Since the softmax assigns the maximum probability to the index with maximum input, we have

$$\arg\max_k p_k^{\text{CLIP}}(x) = \arg\max_k p_k^{\text{Dir}}(x).$$

Thus both distributions yield the same classification decision, proving the accuray invariance.

For calibaration control, observe that $p_k^{\text{Dir}}(x) = e^{s\ell_k} / \sum_j e^{s\ell_j}$ becomes increasingly peaked as $s \to \infty$, converging to a one-hot vector, and tends to the uniform distribution as $s \to 0^+$. The entropy

$$H(p^{\text{Dir}}(x)) = -\sum_k p_k^{\text{Dir}}(x) \log p_k^{\text{Dir}}(x)$$

decreases monotonically with $s$. Thus $s$ leaves classification accuracy unchanged while directly modulating the calibration of predictive confidence.

## E    EXTENDED UNCERTAINTY ANALYSIS

### E.1    IMPLEMENTATION DETAILS FOR UNCERTAINTY QUANTIFICATION

Recall the decomposition of predictive uncertainty under the Dirichlet parameterization into aleatoric uncertainty (AU) and epistemic uncertainty (EU) in Section 3.3.

$$\text{AU}(x) = \mathbb{E}_{\pi \sim \text{Dir}(\alpha)}\big[H(\pi)\big] = -\sum_{k=1}^{C} \frac{\alpha_k}{\alpha_0}\Big(\psi(\alpha_k + 1) - \psi(\alpha_0 + 1)\Big), \quad \text{EU}(x) = \frac{C}{\alpha_0 + C}.$$

Our reformulation $\alpha_k(x) = \exp\big(h(\ell_k^{v \to t}(x))\big)$, $h(\ell) = \frac{\tau\,\ell+1}{\tau'}$, adopts a linear definition of the evidence mapping $h(\ell)$, for which Section 4 and Appendix D have established the theoretical equivalence between CLIP logits and Dirichlet distributions.

In practice, however, the learnable temperature coefficient $\tau$ may become very small during training (e.g., $\tau = 0.01$), which leads to excessively large logits after exponentiation and renders the raw uncertainty values numerically unstable. To address this, we introduce an additional activation $h'(\ell) = \text{softplus}(h(\ell))$, which is commonly adopted in EDL to smooth the outputs and map them into a numerically stable range suitable for analysis Sensoy et al. (2018); Malinin & Gales (2018).

Moreover, when $\tau$ is too small (e.g., $\tau = 0.01$), EU degenerates towards 0 and AU coincides with PU. To avoid this issue, we adopt $\tau = 0.07$ for computing EU, while keeping $\tau = 0.01$ for AU. This choice is theoretically acceptable: both the softplus mapping and the rescaling by $\tau$ affect only the magnitude of uncertainty values, not their ordering. As a result, the reliability of AUROC evaluation, which depends only on ranking, is unaffected. For ECE, we use PU directly computed from probabilities, which is independent of $\tau$ and activation adjustments.

These practical adjustments ensure stable and meaningful AU/EU quantification without altering the comparative reliability of our uncertainty metrics.

### E.2    ADDITIONAL VISUALIZATIONS OF UNCERTAINTY

To complement the main results, we provide extended visualizations of predictive uncertainty under adversarial attacks. Figure 4 reports the degradation of accuracy and predictive uncertainty (PU) across 16 datasets under three strong white-box attacks (PGD, CW, AutoAttack). Figures 5 and 6 further decompose the uncertainty into aleatoric and epistemic components, respectively, comparing CLIP with our method on both clean and adversarial samples. These results illustrate how adversarial perturbations simultaneously reduce accuracy and distort uncertainty, while our method consistently provides more reliable AU/EU estimates across diverse datasets, thereby achieving effective uncertainty calibration.

## F    EVALUATION UNDER LARGER DATASETS, STRONGER ATTACKS, AND ADDITIONAL ABLATIONS

We conduct four extended evaluations to further assess the generality, stability, and architectural robustness of UCAT beyond the main experiments.

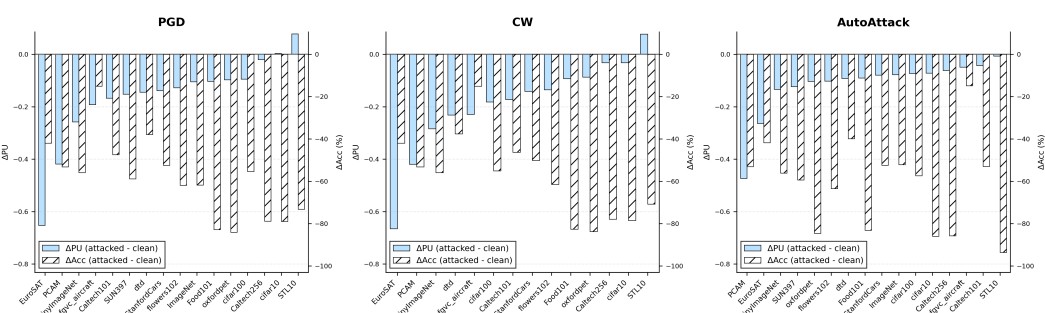

Figure 4: Effect of strong white-box attacks ($\epsilon = 1/255$, 100 steps) on accuracy and predictive uncertainty across 16 datasets. Each panel shows the change under a single attack type (left: PGD, center: CW, right: AutoAttack); for each dataset the filled light bars plot $\Delta \mathrm{PU} = \mathrm{PU}_{\text{attacked}} - \mathrm{PU}_{\text{clean}}$ (left axis) and the hatched bars plot $\Delta \mathrm{Acc} = \mathrm{Acc}_{\text{attacked}} - \mathrm{Acc}_{\text{clean}}$ in percentage points (right axis). Negative values therefore indicate decreases caused by the attack. Results demonstrate that all three attacks induce simultaneous drops in accuracy and predictive uncertainty on most datasets, with the magnitude of degradation varying by dataset and attack.

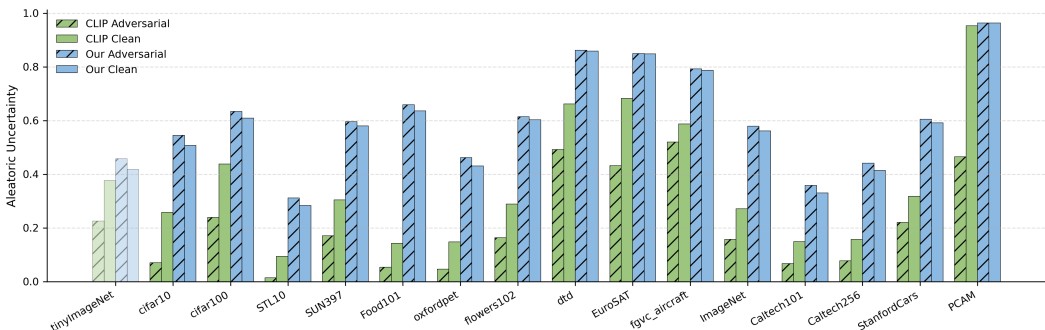

Figure 5: Comparison of **aleatoric uncertainty** on clean and adversarial samples across 16 datasets between CLIP and our method, adversarially trained on tinyImageNet under 10-step PGD with $\epsilon = 2/255$.

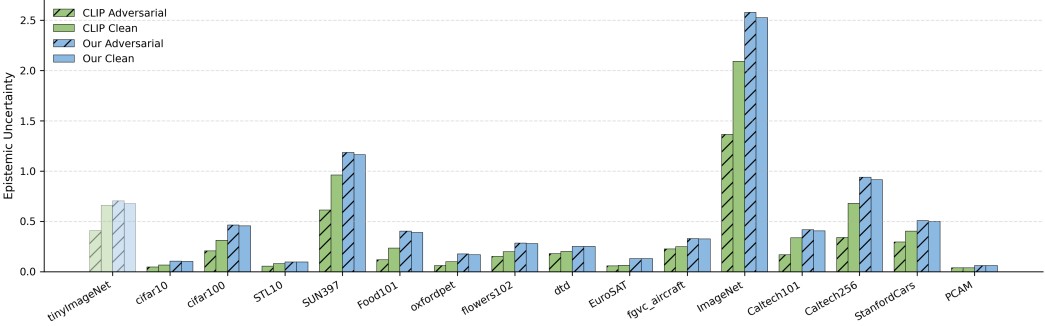

Figure 6: Comparison of **epistemic uncertainty** on clean and adversarial samples across 16 datasets between original CLIP and our method, adversarially trained on tinyImageNet under 10-step PGD with $\epsilon = 2/255$.

### F.1 TRAINING ON A LARGER DATASET

Following TeCoA (Mao et al., 2022), we train on ImageNet-1k with 2-step PGD at $\epsilon = 1/255$ to assess performance on a larger training dataset across 15 benchmarks (tinyImageNet is excluded, as it was not reported in TeCoA's original paper). As shown in Table 4, this setting examines whether UCAT continues to benefit from its uncertainty-calibration mechanism when trained on large-scale data.

Table 4: **Zero-shot adversarial robustness across 15 datasets.** All methods are fine-tuned on ImageNet-1k following TeCoA (Mao et al., 2022), adversarial training uses 2-step PGD (Madry et al., 2017) with $\epsilon = 1/255$. *Average* is the mean across datasets; $H$ is the harmonic mean between Clean and the corresponding robust score. Best and second-best are in **bold** and underline.

| | Methods | Cifar10 | Cifar100 | STL10 | SUN397 | Food101 | Oxfordpets | Flowers102 | DTD | EuroSAT | FGVC Aircraft | ImageNet | Caltech101 | Caltech256 | StanfordCars | PCAM | Average | H |
|---|---|---|---|---|---|---|---|---|---|---|---|---|---|---|---|---|---|---|
| Clean | CLIP (Radford et al., 2021) | 88.03 | 60.45 | 97.03 | 57.26 | 83.89 | 87.41 | 65.49 | 40.64 | 42.66 | 20.16 | 59.15 | 85.32 | 81.73 | 52.02 | 52.08 | 64.89 | |
| | TeCoA (Mao et al., 2022) | 78.12 | 49.68 | 93.30 | 51.28 | 55.37 | 81.58 | 50.92 | 34.15 | 27.57 | 13.89 | 63.87 | 83.51 | 76.51 | 33.30 | 49.01 | 56.14 | |
| | FARE (Schlarmann et al., 2024) | 84.75 | 59.85 | 95.69 | 53.97 | 75.58 | 86.92 | 60.48 | 36.86 | 24.74 | 17.10 | 85.01 | 85.01 | 80.57 | 49.71 | 45.06 | 62.75 | |
| | UCAT (Ours) | 83.78 | 58.11 | 95.65 | 53.98 | 68.84 | 86.05 | 58.30 | 37.18 | 23.02 | 15.24 | 70.48 | 84.64 | 80.27 | 44.96 | 46.57 | 60.47 | |
| Auto Attack | CLIP (Radford et al., 2021) | 9.57 | 4.55 | 35.40 | 1.02 | 3.95 | 2.72 | 1.19 | 2.50 | 0.04 | 0.00 | 1.72 | 24.63 | 7.19 | 0.27 | 0.10 | 0.05 | 0.10 |
| | TeCoA (Mao et al., 2022) | 59.28 | 34.13 | 83.45 | 29.81 | 27.99 | 62.61 | 30.69 | 22.88 | 15.18 | 5.10 | 41.88 | 69.07 | 59.54 | 13.37 | 23.87 | 38.59 | 45.74 |
| | FARE (Schlarmann et al., 2024) | 50.96 | 28.48 | 80.88 | 26.66 | 34.36 | 61.43 | 31.91 | 24.31 | 14.12 | 5.28 | 32.11 | 68.19 | 59.95 | 18.52 | 25.74 | 37.53 | 46.97 |
| | UCAT (Ours) | 50.59 | 28.48 | 82.09 | 29.93 | 33.72 | 67.59 | 33.26 | 24.42 | 12.65 | 5.73 | 47.51 | 71.11 | 62.71 | 19.62 | 25.84 | 39.68 | 47.92 |
| PGD | CLIP (Radford et al., 2021) | 2.54 | 1.11 | 3.18 | 0.05 | 0.03 | 0.03 | 0.02 | 0.19 | 0.17 | 0.23 | 0.04 | 0.10 | 0.26 | 0.07 | 0.12 | 0.54 | 1.08 |
| | TeCoA (Mao et al., 2022) | 58.27 | 32.57 | 83.16 | 29.03 | 25.79 | 61.76 | 28.93 | 20.70 | 13.26 | 4.05 | 48.51 | 68.40 | 58.59 | 12.03 | 24.09 | 37.94 | 45.28 |
| | FARE (Schlarmann et al., 2024) | 49.62 | 25.98 | 80.60 | 24.77 | 33.06 | 60.51 | 29.55 | 22.02 | 12.95 | 4.08 | 39.81 | 67.21 | 58.87 | 16.43 | 25.56 | 36.73 | 46.34 |
| | UCAT (Ours) | 49.00 | 26.42 | 81.73 | 27.85 | 31.88 | 66.86 | 30.64 | 22.45 | 10.76 | 4.50 | 45.59 | 70.12 | 61.64 | 17.40 | 25.37 | 38.15 | 46.78 |

### F.2 TRAINING UNDER STRONGER ADVERSARIAL ATTACKS

TeCoA (Mao et al., 2022) and FARE (Schlarmann et al., 2024) represent two widely used adversarial fine-tuning configurations on TinyImageNet, with FARE adopting a substantially stronger perturbation budget. To ensure a fair and comprehensive comparison, we evaluate UCAT under this stronger training regime as well. Specifically, we follow the FARE configuration and train models using 10-step PGD with $\epsilon = 2/255$.

Table 5: **Zero-shot adversarial robustness under 10-step PGD training.** All methods are fine-tuned on TinyImageNet following FARE (Schlarmann et al., 2024), adversarial training uses 10-step PGD (Madry et al., 2017) with $\epsilon = 2/255$. *Average* is the mean across datasets; $H$ is the harmonic mean between Clean and the corresponding robust score. Best and second-best are in **bold** and underline.

| | Methods | TinyImageNet | Cifar10 | Cifar100 | STL10 | SUN397 | Food101 | Oxfordpets | Flowers102 | DTD | EuroSAT | FGVC Aircraft | ImageNet | Caltech101 | Caltech256 | StanfordCars | PCAM | Average | H |
|---|---|---|---|---|---|---|---|---|---|---|---|---|---|---|---|---|---|---|---|
| Clean | CLIP (Radford et al., 2021) | 57.96 | 88.03 | 60.45 | 97.03 | 57.26 | 83.89 | 87.41 | 65.49 | 40.64 | 42.66 | 20.16 | 59.15 | 85.32 | 81.73 | 52.02 | 52.08 | 64.45 | |
| | TeCoA (Mao et al., 2022) | 63.20 | 58.62 | 31.75 | 80.59 | 25.71 | 19.15 | 49.25 | 24.61 | 17.34 | 15.89 | 2.88 | 24.70 | 63.04 | 47.67 | 13.11 | 49.97 | 36.72 | |
| | FARE (Schlarmann et al., 2024) | 16.92 | 40.23 | 11.96 | 64.56 | 7.89 | 8.07 | 19.24 | 11.82 | 7.93 | 12.52 | 2.55 | 7.98 | 47.14 | 27.61 | 6.06 | 50.02 | 21.41 | |
| | PMG-AFT (Wang et al., 2024) | 22.16 | 74.05 | 33.81 | 92.75 | 55.66 | 72.69 | 83.10 | 55.86 | 28.30 | 19.83 | 17.46 | 51.46 | 80.83 | 75.22 | 43.09 | 48.72 | 53.44 | |
| | TGA-ZSR (Yu et al., 2024) | 69.78 | 83.98 | 52.32 | 91.36 | 44.70 | 50.17 | 72.55 | 45.05 | 26.92 | 27.58 | 10.68 | 41.84 | 80.04 | 71.94 | 33.14 | 50.02 | 53.25 | |
| | UCAT (Ours) | 67.18 | 74.05 | 41.07 | 86.73 | 30.10 | 36.97 | 62.66 | 36.69 | 24.95 | 19.39 | 7.26 | 32.61 | 75.00 | 60.15 | 26.39 | 49.66 | 45.21 | |
| Auto Attack | CLIP (Radford et al., 2021) | 0.00 | 0.05 | 0.14 | 0.00 | 0.02 | 0.03 | 0.03 | 0.02 | 0.13 | 0.17 | 0.23 | 0.03 | 0.02 | 0.06 | 0.07 | 0.12 | 0.07 | 0.14 |
| | TeCoA (Mao et al., 2022) | 32.68 | 21.94 | 13.17 | 51.93 | 8.43 | 5.54 | 21.56 | 9.92 | 9.52 | 11.36 | 0.51 | 9.10 | 38.88 | 25.35 | 2.56 | 49.23 | 19.48 | 25.46 |
| | FARE (Schlarmann et al., 2024) | 7.00 | 14.81 | 4.58 | 38.71 | 2.81 | 2.16 | 6.49 | 4.29 | 4.47 | 8.52 | 0.72 | 2.86 | 29.84 | 14.03 | 1.34 | 50.02 | 12.04 | 15.41 |
| | PMG-AFT (Wang et al., 2024) | 0.00 | 0.96 | 0.74 | 0.60 | 0.05 | 0.06 | 0.06 | 0.05 | 0.27 | 0.03 | 0.04 | 0.04 | 0.92 | 0.26 | 0.04 | 0.21 | 0.25 | 0.51 |
| | TGA-ZSR (Yu et al., 2024) | 11.28 | 6.29 | 5.53 | 36.76 | 4.00 | 3.27 | 9.27 | 6.18 | 6.33 | 8.94 | 0.18 | 5.13 | 30.23 | 19.57 | 0.96 | 42.88 | 12.30 | 19.98 |
| | UCAT (Ours) | 32.84 | 24.08 | 13.97 | 57.15 | 9.50 | 9.09 | 24.72 | 12.60 | 11.97 | 3.76 | 0.78 | 11.11 | 49.44 | 32.35 | 4.71 | 32.62 | 20.67 | 28.37 |
| CW | CLIP (Radford et al., 2021) | 0.00 | 0.58 | 0.20 | 0.57 | 0.08 | 0.00 | 0.00 | 0.00 | 0.12 | 0.00 | 0.00 | 0.08 | 0.24 | 0.33 | 2.19 | 0.00 | 0.05 | 0.55 |
| | TeCoA (Mao et al., 2022) | 33.90 | 23.05 | 13.66 | 52.50 | 8.95 | 5.74 | 22.13 | 10.05 | 9.42 | 11.40 | 0.63 | 9.58 | 39.92 | 26.04 | 3.10 | 49.26 | 19.96 | 25.86 |
| | FARE (Schlarmann et al., 2024) | 7.04 | 14.64 | 4.63 | 38.94 | 2.91 | 2.20 | 6.68 | 4.38 | 4.36 | 8.43 | 0.75 | 2.96 | 30.13 | 14.24 | 1.73 | 50.02 | 12.13 | 15.48 |
| | PMG-AFT (Wang et al., 2024) | 0.02 | 1.63 | 0.70 | 2.73 | 0.09 | 0.02 | 0.03 | 0.00 | 0.32 | 0.00 | 0.00 | 0.11 | 3.03 | 1.13 | 1.87 | 0.00 | 0.73 | 1.44 |
| | TGA-ZSR (Yu et al., 2024) | 29.68 | 20.48 | 11.48 | 51.53 | 9.15 | 6.43 | 21.72 | 12.05 | 9.63 | 11.01 | 0.60 | 10.06 | 28.67 | | 4.48 | 49.97 | 19.86 | 28.93 |
| | UCAT (Ours) | 34.64 | 25.46 | 14.69 | 57.88 | 10.25 | 9.83 | 26.49 | 12.91 | 11.70 | 3.80 | 1.11 | 11.95 | 50.47 | 33.40 | 6.36 | 33.09 | 21.50 | 29.14 |
| PGD | CLIP (Radford et al., 2021) | 0.00 | 0.94 | 0.28 | 0.45 | 0.00 | 0.00 | 0.00 | 0.00 | 0.11 | 0.00 | 0.00 | 0.00 | 0.77 | 0.19 | 0.00 | 0.00 | 0.00 | 0.00 |
| | TeCoA (Mao et al., 2022) | 35.74 | 23.57 | 14.47 | 53.50 | 10.20 | 6.58 | 22.90 | 10.96 | 10.75 | 11.72 | 0.57 | 10.53 | 40.27 | 27.02 | 3.61 | 49.31 | 20.73 | 26.50 |
| | FARE (Schlarmann et al., 2024) | 8.62 | 18.19 | 5.67 | 41.10 | 3.46 | 2.88 | 7.01 | 5.22 | 5.21 | 9.29 | 0.96 | 3.47 | 31.98 | 15.47 | 1.73 | 50.02 | 13.14 | 16.29 |
| | PMG-AFT (Wang et al., 2024) | 0.12 | 24.10 | 3.76 | 16.34 | 0.16 | 0.04 | 0.30 | 0.23 | 2.29 | 0.50 | 0.00 | 0.23 | 4.53 | 2.02 | 0.01 | 47.90 | 6.41 | 11.44 |
| | TGA-ZSR (Yu et al., 2024) | 30.74 | 20.17 | 12.02 | 51.99 | 9.46 | 6.69 | 20.58 | 12.47 | 10.85 | 11.22 | 0.63 | 10.28 | 40.63 | 29.06 | 3.56 | 49.97 | 20.02 | 29.10 |
| | UCAT (Ours) | 35.38 | 25.81 | 15.67 | 58.44 | 11.48 | 11.17 | 26.82 | 15.04 | 13.94 | 4.53 | 1.20 | 13.13 | 51.34 | 34.60 | 6.72 | 34.02 | 22.45 | 30.01 |

We first report zero-shot robustness under AutoAttack in Table 5, which measures performance against substantially stronger perturbations than those used in Table 2. To further assess generalization across attack types, we additionally evaluate CAA (Mao et al., 2021) and A³ (Liu et al., 2022),

two strong adaptive attacks (Table 6). Together, these evaluations examine UCAT's robustness under both stronger training perturbations and a wider range of test-time threat models.

Table 6: **Performance of all methods under stronger adversarial attacks, CAA (Mao et al., 2021) and A[3] (Liu et al., 2022).** All methods are fine-tuned on TinyImageNet following FARE (Schlarmann et al., 2024), adversarial training uses 10-step PGD (Madry et al., 2017) with $\epsilon = 2/255$. *Average* is the mean across datasets. $H$ is the harmonic mean between Clean and the corresponding robust score. Best and second-best are in **bold** and underline.

| | Methods | TinyImageNet | Cifar10 | Cifar100 | STL10 | SUN397 | Food101 | Oxfordpets | Flowers102 | DTD | EuroSAT | FGVC Aircraft | ImageNet | Caltech101 | Caltech256 | StanfordCars | PCAM | Average | H |
|---|---|---|---|---|---|---|---|---|---|---|---|---|---|---|---|---|---|---|---|
| Clean | CLIP (Radford et al., 2021) | 57.96 | 88.02 | 60.47 | 97.03 | 57.26 | 83.89 | 87.38 | 65.52 | 40.69 | 42.65 | 20.16 | 59.15 | 85.33 | 81.73 | 51.98 | 52.08 | 64.46 | |
| | **TeCoA (Mao et al., 2022)** | 63.20 | 58.62 | 31.75 | 80.59 | 25.71 | 19.15 | 49.25 | 24.61 | 17.34 | 15.89 | 2.88 | 24.70 | 63.04 | 47.67 | 13.11 | 49.97 | 36.72 | |
| | FARE (Schlarmann et al., 2024) | 16.92 | 40.23 | 11.96 | 64.56 | 7.89 | 8.07 | 19.24 | 11.82 | 7.93 | 12.52 | 2.55 | 7.98 | 47.14 | 27.61 | 6.06 | 50.02 | 21.41 | |
| | PMG-AFT (Wang et al., 2024) | 46.62 | 70.37 | 38.49 | 90.34 | 52.02 | 57.11 | 79.75 | 49.36 | 32.23 | 23.43 | 12.03 | 47.70 | 82.49 | 73.65 | 41.60 | 55.87 | 53.32 | |
| | TGA-ZSR (Yu et al., 2024) | 69.78 | 83.98 | 52.32 | 91.36 | 44.70 | 50.17 | 72.55 | 45.05 | 26.92 | 27.58 | 10.68 | 41.84 | 80.04 | 71.94 | 33.14 | 50.02 | 53.25 | |
| | UCAT | 67.18 | 66.52 | 41.07 | 86.73 | 30.10 | 36.97 | 62.66 | 36.69 | 24.95 | 19.39 | 7.26 | 32.61 | 75.00 | 60.15 | 26.39 | 49.66 | 45.21 | |
| CAA | CLIP (Radford et al., 2021) | 1.90 | 5.84 | 0.32 | 28.99 | 0.63 | 9.80 | 5.04 | 1.29 | 0.05 | 0.04 | 0.00 | 1.02 | 16.63 | 13.17 | 0.42 | 0.00 | 5.32 | 9.83 |
| | TeCoA (Mao et al., 2022) | 35.52 | 23.40 | 14.38 | 53.25 | 9.98 | 6.49 | 22.65 | 10.95 | 10.69 | 11.62 | 0.57 | 10.33 | 40.03 | 26.83 | 3.43 | 49.28 | 20.59 | 26.38 |
| | FARE (Schlarmann et al., 2024) | 8.50 | 17.74 | 5.51 | 40.86 | 3.39 | 2.80 | 6.95 | 5.14 | 5.05 | 8.86 | 0.93 | 3.41 | 31.72 | 15.32 | 1.65 | 50.02 | 12.99 | 16.17 |
| | PMG-AFT (Wang et al., 2024) | 7.18 | 13.18 | 3.79 | 56.18 | 3.37 | 8.21 | 13.63 | 4.36 | 3.19 | 0.32 | 0.00 | 4.43 | 39.61 | 26.36 | 0.44 | 0.39 | 11.54 | 18.97 |
| | TGA-ZSR (Yu et al., 2024) | 18.64 | 10.77 | 8.41 | 42.23 | 5.78 | 4.51 | 11.72 | 7.81 | 7.87 | 10.54 | 0.27 | 6.73 | 33.71 | 22.49 | 1.62 | 42.37 | 14.72 | 23.06 |
| | **UCAT** | 35.12 | 25.69 | 15.42 | 58.33 | 11.16 | 10.91 | 26.90 | 14.73 | 13.83 | 4.33 | 1.08 | 12.75 | 51.11 | 34.33 | 6.29 | 33.45 | 22.21 | 29.79 |
| A[3] | CLIP (Radford et al., 2021) | 1.26 | 6.47 | 0.33 | 30.70 | 0.70 | 9.73 | 4.80 | 1.11 | 0.11 | 0.10 | 0.00 | 1.00 | 19.11 | 13.56 | 0.37 | 0.00 | 5.58 | 10.28 |
| | TeCoA (Mao et al., 2022) | 35.38 | 23.21 | 14.22 | 53.11 | 9.95 | 6.46 | 22.57 | 10.83 | 10.59 | 11.62 | 0.57 | 10.29 | 39.98 | 26.75 | 3.47 | 49.27 | 20.52 | 26.32 |
| | FARE (Schlarmann et al., 2024) | 8.46 | 17.55 | 5.49 | 40.65 | 3.39 | 2.78 | 6.87 | 5.12 | 5.05 | 8.89 | 0.93 | 3.40 | 31.69 | 15.31 | 1.65 | 50.02 | 12.95 | 16.14 |
| | PMG-AFT (Wang et al., 2024) | 7.02 | 13.16 | 3.65 | 55.74 | 3.37 | 7.90 | 12.87 | 4.68 | 4.95 | 4.45 | 0.00 | 4.37 | 39.60 | 26.18 | 0.44 | 0.89 | 11.83 | 19.36 |
| | TGA-ZSR (Yu et al., 2024) | 30.02 | 19.54 | 11.59 | 51.60 | 9.02 | 6.44 | 19.98 | 12.21 | 10.64 | 11.13 | 0.54 | 9.85 | 40.07 | 28.50 | 3.20 | 49.95 | 19.64 | 28.70 |
| | UCAT | 34.90 | 25.44 | 15.33 | 58.14 | 11.08 | 10.82 | 26.41 | 14.67 | 13.83 | 4.33 | 1.08 | 12.69 | 51.02 | 34.22 | 6.19 | 33.34 | 22.09 | 29.68 |

## F.3 EFFECT OF THE CALIBRATION COEFFICIENT $\tau'$

As shown in Table 7, We perform a controlled ablation by varying the Dirichlet calibration coefficient $\tau'$ and reporting clean accuracy, AutoAttack robustness, and their harmonic mean $H$ across 16 datasets. This study isolates the influence of the evidence-scaling term in our formulation and identifies $\tau' = 0.07$ as the best operating point.

Table 7: **Effect of the calibration coefficient $\tau'$ on zero-shot adversarial robustness across 16 single-label datasets.** All methods are fine-tuned on TinyImageNet following TeCoA (Yu et al., 2024), adversarial training uses 2-step PGD (Madry et al., 2017) with $\epsilon = 1/255$. *Average* is the mean across datasets. $H$ is the harmonic mean between Clean and the corresponding robust score. Best and second-best are in **bold** and underline.

| | Methods | TinyImageNet | Cifar10 | Cifar100 | STL10 | SUN397 | Food101 | Oxfordpets | Flowers102 | DTD | EuroSAT | FGVC Aircraft | ImageNet | Caltech101 | Caltech256 | StanfordCars | PCAM | Average | H |
|---|---|---|---|---|---|---|---|---|---|---|---|---|---|---|---|---|---|---|---|
| Clean | CLIP (Radford et al., 2021) | 57.96 | 88.02 | 60.47 | 97.03 | 57.26 | 83.89 | 87.38 | 65.52 | 40.69 | 42.65 | 20.16 | 59.15 | 85.33 | 81.73 | 51.98 | 52.08 | 64.46 | |
| | $\tau' = 0.01$ | 70.06 | 70.33 | 39.38 | 86.74 | 37.22 | 31.12 | 63.78 | 35.16 | 25.43 | 16.87 | 4.89 | 34.23 | 73.42 | 60.46 | 20.56 | 49.99 | 44.98 | |
| | $\tau' = 0.03$ | 69.38 | 64.18 | 36.23 | 86.56 | 36.74 | 31.13 | 62.50 | 35.52 | 25.75 | 14.24 | 5.13 | 33.86 | 73.30 | 59.63 | 20.15 | 49.92 | 44.01 | |
| | $\tau' = 0.05$ | 72.68 | 71.18 | 42.52 | 88.33 | 36.36 | 33.91 | 66.78 | 37.10 | 26.44 | 16.86 | 5.85 | 35.98 | 76.29 | 61.66 | 24.69 | 49.83 | 46.65 | |
| | $\tau' = 0.07$ **(Ours)** | 74.46 | 81.81 | 54.45 | 91.88 | 41.06 | 53.58 | 74.16 | 47.57 | 31.92 | 19.29 | 10.95 | 43.20 | 82.39 | 71.53 | 37.32 | 51.20 | 54.17 | |
| | $\tau' = 0.09$ | 70.66 | 83.79 | 56.44 | 92.23 | 42.10 | 58.26 | 75.88 | 49.16 | 33.56 | 16.87 | 11.67 | 43.62 | 82.07 | 72.96 | 38.48 | 56.23 | 55.25 | |
| | $\tau' = 0.10$ | 70.40 | 85.27 | 57.27 | 92.16 | 42.51 | 57.85 | 76.34 | 48.97 | 32.93 | 15.28 | 11.19 | 43.56 | 81.78 | 72.90 | 38.43 | 56.08 | 55.18 | |
| Auto Attack | CLIP (Radford et al., 2021) | 1.26 | 6.47 | 0.33 | 30.70 | 0.70 | 9.73 | 4.80 | 1.11 | 0.11 | 0.10 | 0.00 | 1.00 | 19.11 | 13.56 | 0.37 | 0.00 | 5.58 | 10.28 |
| | $\tau' = 0.01$ | 45.98 | 37.00 | 20.52 | 69.35 | 18.07 | 12.52 | 37.80 | 20.31 | 16.86 | 11.53 | 1.74 | 17.21 | 54.62 | 41.57 | 7.82 | 49.16 | 28.89 | 35.18 |
| | $\tau' = 0.03$ | 46.30 | 33.71 | 18.71 | 68.74 | 18.08 | 13.20 | 37.78 | 20.61 | 16.70 | 11.29 | 1.77 | 17.41 | 55.72 | 41.57 | 7.46 | 46.95 | 28.50 | 34.60 |
| | $\tau' = 0.05$ | 48.48 | 37.96 | 21.59 | 70.98 | 17.72 | 14.17 | 40.07 | 20.26 | 17.23 | 11.42 | 1.44 | 18.35 | 58.42 | 42.95 | 9.59 | 45.69 | 29.77 | 36.35 |
| | $\tau' = 0.07$ **(Ours)** | 45.80 | 42.32 | 23.03 | 73.15 | 18.26 | 20.52 | 44.02 | 24.54 | 18.14 | 2.26 | 2.61 | 20.15 | 63.73 | 48.66 | 12.60 | 29.51 | 30.58 | 39.09 |
| | $\tau' = 0.09$ | 38.58 | 41.00 | 21.23 | 71.83 | 16.93 | 21.60 | 40.67 | 25.08 | 17.87 | 1.58 | 2.43 | 18.98 | 63.36 | 47.54 | 11.37 | 23.08 | 28.94 | 37.99 |
| | $\tau' = 0.10$ | 37.64 | 40.75 | 19.94 | 70.90 | 16.06 | 20.57 | 40.77 | 24.15 | 17.18 | 1.24 | 2.28 | 18.04 | 61.53 | 46.28 | 10.96 | 17.82 | 27.88 | 37.04 |

## F.4 GENERALIZATION ACROSS VISION–LANGUAGE MODELS

We further examine whether UCAT is tied to the CLIP-B/32 backbone. We fine-tune SLIP-B16 (Mu et al., 2022), CLIP-B/16 (Radford et al., 2021), and CLIP-B/32 (Radford et al., 2021) on TinyImageNet using the same zero-shot adversarial robustness (ZSAR) configuration (2-step PGD, $\epsilon = 1/255$) in Table 8. This evaluates the architecture-agnostic nature of our Dirichlet alignment.

## F.5 COMPARISON WITH CLASSICAL ADVERSARIAL TRAINING BASELINES

Table 8: **Zero-shot adversarial robustness on different vision–language models (VLMs).** All methods are fine-tuned on TinyImageNet following TeCoA (Yu et al., 2024), adversarial training uses 2-step PGD (Madry et al., 2017) with $\epsilon = 1/255$. *Average* is the mean across datasets. *H* is the harmonic mean between Clean and the corresponding robust score. Best and second-best are in **bold** and underline.

| Methods | | TinyImageNet | Cifar10 | Cifar100 | STL10 | SUN397 | Food101 | Oxfordpets | Flowers102 | DTD | EuroSAT | FGVC Aircraft | ImageNet | Caltech101 | Caltech256 | StanfordCars | PCAM | Average | H |
|---|---|---|---|---|---|---|---|---|---|---|---|---|---|---|---|---|---|---|---|
| CLIP-B/16 (Radford et al., 2021) | Clean | 60.86 | 89.49 | 66.25 | 98.03 | 61.05 | 88.51 | 88.66 | 70.97 | 43.03 | 45.83 | 24.57 | 0.07 | 86.07 | 84.99 | 58.29 | 52.86 | 63.72 | |
| | AutoAttack | 0.00 | 0.00 | 0.03 | 0.00 | 0.01 | 0.00 | 0.00 | 0.02 | 0.00 | 0.08 | 0.06 | 0.00 | 0.01 | 0.00 | 0.03 | 0.00 | 0.01 | 0.03 |
| + UCAT (Ours) | Clean | 77.52 | 82.02 | 57.34 | 92.66 | 43.78 | 54.76 | 73.75 | 49.86 | 30.16 | 24.25 | 12.39 | 0.06 | 82.21 | 73.72 | 37.51 | 54.50 | 52.91 | |
| | AutoAttack | 50.90 | 46.15 | 26.05 | 76.83 | 19.74 | 21.56 | 45.08 | 27.13 | 17.66 | 2.37 | 3.84 | 0.00 | 65.09 | 52.64 | 13.02 | 20.65 | 30.54 | 38.73 |
| SLIP-B/16 (Mu et al., 2022) | Clean | 36.74 | 78.97 | 44.22 | 94.28 | 52.71 | 59.70 | 31.67 | 59.67 | 21.65 | 19.87 | 5.79 | 38.61 | 75.15 | 62.71 | 5.85 | 48.96 | 46.03 | |
| | AutoAttack | 0.04 | 0.00 | 0.02 | 0.01 | 0.02 | 0.03 | 0.00 | 0.00 | 0.11 | 0.04 | 0.00 | 0.02 | 0.04 | 0.03 | 0.00 | 0.01 | 0.02 | 0.05 |
| + UCAT (Ours) | Clean | 51.90 | 70.14 | 38.51 | 86.76 | 41.86 | 26.59 | 25.29 | 38.19 | 16.22 | 13.50 | 3.75 | 26.42 | 70.12 | 51.07 | 3.37 | 50.29 | 38.37 | |
| | AutoAttack | 25.52 | 34.97 | 16.04 | 66.73 | 19.13 | 8.77 | 7.39 | 16.34 | 8.09 | 1.23 | 0.75 | 11.58 | 49.03 | 29.35 | 0.57 | 30.92 | 20.40 | 0.10 |
| CLIP-B/32 (Radford et al., 2021) | Clean | 57.27 | 88.05 | 60.47 | 97.04 | 57.27 | 83.89 | 87.35 | 65.52 | 40.80 | 42.50 | 20.13 | 59.15 | 85.33 | 81.72 | 52.02 | 52.24 | 64.42 | |
| | AutoAttack | 1.26 | 6.47 | 0.33 | 30.70 | 0.70 | 9.73 | 4.80 | 1.11 | 0.11 | 0.10 | 0.00 | 1.00 | 19.11 | 13.56 | 0.37 | 0.00 | 5.58 | 10.28 |
| + UCAT (Ours) | Clean | 74.46 | 81.81 | 54.45 | 91.88 | 41.06 | 53.58 | 74.16 | 47.57 | 31.92 | 19.29 | 10.95 | 43.20 | 82.39 | 71.53 | 37.32 | 51.20 | 54.17 | |
| | AutoAttack | 45.80 | 42.32 | 23.03 | 73.15 | 18.26 | 20.52 | 44.02 | 24.54 | 18.14 | 2.26 | 2.61 | 20.15 | 63.73 | 48.66 | 12.60 | 29.51 | 30.58 | 39.09 |

Finally, we examine whether classical adversarial training methods (AT) can serve as a baseline for zero-shot adversarial robustness (ZSAR). For fairness, all AT baselines (TRADES (Zhang et al., 2019), ACAT (Addepalli et al., 2022), DKL (Cui et al., 2024)) are fine-tuned on TinyImageNet using the same 2-step PGD configuration ($\epsilon = 1/255$) with their *native objective, supervision signal, and optimization hyperparameters*. These models are then evaluated in a zero-shot manner on downstream ZSAR benchmarks, where no task-specific labels are used during testing. Although these AT methods were not originally developed for zero-shot scenarios, they still achieve strong performance under ZSAR (Table 9), demonstrating the general effectiveness of their theoretical formulations. However, compared with these supervised AT approaches, our UCAT method, which is explicitly designed to preserve open-set semantic alignment, achieves consistently better overall ZSAR performance, particularly in terms of the harmonic mean between clean and robust accuracy.

Table 9: **Comparison with representative adversarial training methods zero-shot adversarial robustness (ZSAR) setting.** All methods are fine-tuned on TinyImageNet following TeCoA (Yu et al., 2024), adversarial training uses 2-step PGD (Madry et al., 2017) with $\epsilon = 1/255$. *Average* is the mean across datasets. *H* is the harmonic mean between Clean and the corresponding robust score. Best and second-best are in **bold** and underline.

| | Methods | TinyImageNet | Cifar10 | Cifar100 | STL10 | SUN397 | Food101 | Oxfordpets | Flowers102 | DTD | EuroSAT | FGVC Aircraft | ImageNet | Caltech101 | Caltech256 | StanfordCars | PCAM | Average | H |
|---|---|---|---|---|---|---|---|---|---|---|---|---|---|---|---|---|---|---|---|
| Clean | CLIP (Radford et al., 2021) | 57.96 | 88.02 | 60.47 | 97.03 | 57.26 | 83.89 | 87.38 | 65.52 | 40.69 | 42.65 | 20.16 | 59.15 | 85.33 | 81.73 | 51.98 | 52.08 | 64.46 | |
| | TRADES (Zhang et al., 2019) | 67.73 | 62.05 | 34.38 | 80.81 | 26.64 | 22.68 | 54.43 | 24.48 | 20.59 | 16.51 | 3.60 | 26.31 | 63.77 | 50.89 | 14.59 | 49.95 | 38.71 | |
| | ACAT (Addepalli et al., 2022) | 72.80 | 64.72 | 34.71 | 82.41 | 30.06 | 19.13 | 60.40 | 26.80 | 17.29 | 15.62 | 4.41 | 28.60 | 64.94 | 50.89 | 15.82 | 50.01 | 39.91 | |
| | DKL (Cui et al., 2024) | 70.84 | 65.31 | 35.61 | 82.54 | 30.11 | 21.11 | 55.11 | 25.94 | 21.17 | 16.26 | 3.96 | 27.18 | 65.84 | 52.00 | 14.20 | 48.97 | 39.76 | |
| | UCAT (Ours) | 74.46 | 81.81 | 54.45 | 91.88 | 41.06 | 53.58 | 74.16 | 47.57 | 31.92 | 19.29 | 10.95 | 43.20 | 82.39 | 71.53 | 37.32 | 51.20 | 54.17 | |
| AutoAttack | CLIP (Radford et al., 2021) | 1.26 | 6.47 | 0.33 | 30.70 | 0.70 | 9.73 | 4.80 | 1.11 | 0.11 | 0.10 | 0.00 | 1.00 | 19.11 | 13.56 | 0.37 | 0.00 | 5.58 | 10.28 |
| | TRADES (Zhang et al., 2019) | 52.19 | 41.39 | 22.44 | 67.59 | 15.69 | 12.50 | 37.42 | 16.46 | 15.75 | 11.77 | 1.23 | 16.16 | 51.16 | 37.84 | 7.29 | 46.42 | 28.33 | 32.72 |
| | ACAT (Addepalli et al., 2022) | 51.85 | 34.54 | 16.85 | 64.84 | 14.99 | 8.95 | 36.71 | 16.56 | 10.96 | 11.41 | 1.59 | 15.69 | 48.70 | 34.84 | 7.62 | 49.89 | 26.62 | 31.94 |
| | DKL (Cui et al., 2024) | 54.71 | 41.77 | 22.29 | 67.84 | 17.32 | 10.73 | 36.06 | 16.23 | 15.69 | 11.72 | 1.65 | 16.66 | 51.85 | 38.33 | 6.67 | 43.48 | 28.31 | 33.07 |
| | UCAT (Ours) | 45.80 | 42.32 | 23.03 | 73.15 | 18.26 | 20.52 | 44.02 | 24.54 | 18.14 | 2.26 | 2.61 | 20.15 | 63.73 | 48.66 | 12.60 | 29.51 | 30.58 | 39.09 |

# G LIMITATIONS AND FUTURE WORK

While our study is focused on a specific setting, it highlights several opportunities for future exploration. First, in the current setting we only consider adversarial perturbations applied to the image encoder, while future work may extend to more comprehensive bidirectional attacks that also target the text encoder. Second, our framework requires fine-tuning, whereas recent work has explored test-time defenses based on prior assumptions without additional training (Xing et al., 2025; Zhang et al., 2025). However, such approaches often show instability under adaptive attacks such as AutoAttack. Incorporating our uncertainty-based analysis as a principled prior into test-time defenses is a promising future direction. Finally, our experiments are restricted to CLIP, and it will be valuable to investigate the applicability of our Dirichlet-based uncertainty calibration to larger and more diverse vision–language models.

