# OpenReview forum: "Calibrating Uncertainty for Zero-Shot Adversarial CLIP"
_ICLR.cc/2026/Conference — Submitted to ICLR 2026_

### Official Review · Reviewer_of3Z · 2025-10-19

**Soundness:** 4
**Presentation:** 4
**Contribution:** 4
**Rating:** 8
**Confidence:** 3

**Summary:**

The paper addresses the issue that adversarial perturbations not only degrade accuracy but also suppress uncertainty, leading to severe miscalibration and unreliable overconfidence. This overlooked phenomenon highlights a critical reliability gap beyond robustness. The authors reformulate CLIP’s logits as concentration parameters of a Dirichlet distribution and propose a novel uncertainty-calibrated adversarial fine-tuning method that regularizes entire Dirichlet distributions to jointly preserve inter-class relations and calibrate evidence strength. The results demonstrate that the proposed method effectively calibrates uncertainty under attack while maintaining strong clean accuracy and competitive adversarial robustness.

**Strengths:**

1. Well-motivated research question: The paper addresses an important and overlooked issue regarding the impact of adversarial perturbations on uncertainty calibration. The study is generally well-organized and clearly motivated.

2. Technical soundness: Reformulating CLIP’s logits as concentration parameters of a Dirichlet distribution, coupled with the proposed uncertainty-calibrated adversarial fine-tuning method, is technically sound and well-executed.

3. Comprehensive evaluation: The evaluation is thorough, covering different types of attacks, varying attack strengths, and multiple datasets. Additionally, aablation studies and analyses of key hyperparameters are provided.

**Weaknesses:**

1. A key concern is that the method is evaluated on only a single variant of CLIP—CLIP-B/32. It would be valuable to see results on additional CLIP variants or applications to other VLMs better demonstrate generalizability.
2. Could the authors provide some explanations for the comparison between FARE2 and the proposed method in the results of Figure 3(b)? Specifically, why does FARE2 achieve better uncertainty calibration but worse robustness accuracy, while the proposed method strikes a better balance?

**Questions:**

See weaknesses above.

---

> ### Author Response · Authors · 2025-11-21
> **Response to Reviewer of3Z [1/1]**
>
> Dear Reviewer of3Z, thank you very much for your positive and insightful feedback. We are grateful for your thoughtful comments. Below we provide detailed responses, and we will integrate the corresponding clarifications and improvements into the updated version of the paper.
>
> ---
>
> ## W1
>
> Thank you for the suggestion. We agree that evaluating additional CLIP variants is important for demonstrating generalizability. Our initial choice of **CLIP-B/32** followed prior zero-shot adversarial robustness work such as TeCoA, but our method is **not tied to this architecture**. UCAT (our Uncertainty-Calibrated Adversarial fine-Tuning method) operates by aligning CLIP’s contrastive similarity logits with a Dirichlet distribution, and this formulation applies broadly to **any model trained with InfoNCE or similar contrastive losses**.
>
> To verify this, we evaluated several additional vision–language models under the same protocol as Table 2 (Trained on TinyImageNet with 2-step PGD and evaluated under AutoAttack with $\epsilon = 1/255$). We tested: **CLIP-B/16** and **SLIP-B/16** (InfoNCE + SimCLR). For each model, we report both the original version and the version fine-tuned with UCAT, and reported the average results across all 16 datasets.
>
> | Model      | Clean     | AutoAttack | Harmonic  |
> | ---------- | --------- | ---------- | --------- |
> | CLIP-B/16  | 63.72     | 0.01       | 0.03      |
> | **+ UCAT** | **52.91** | **30.54**  | **38.73** |
> | SLIP-B/16  | 46.03     | 0.02       | 0.05      |
> | **+ UCAT** | **38.37** | **20.40**  | **26.64** |
> | CLIP-B/32  | 64.42     | 0.51       | 1.01      |
> | **+ UCAT** | **54.17** | **30.58**  | **39.09** |
>
> Across all architectures, other CLIP variant and contrastive-augmented SLIP, UCAT consistently improves adversarial accuracy and harmonic robustness. These results demonstrate that our method is **model-agnostic**, broadly applicable to **multiple CLIP variants and other contrastive VLMs**, and not restricted to CLIP-B/32.
>
> ---
>
> ## W2
>
> Thank you for the question. The different behaviors of FARE2 and our UCAT mainly come from the **different alignment targets used to achieve adversarial robustness**.
>
> **Why FARE2 has lower ECE but weaker robustness.** FARE aligns adversarial *image features* to their clean counterparts. This feature-level alignment keeps the perturbed embedding close to the clean one in magnitude, but it does **not** recover the correct *semantic direction* required in zero-shot CLIP. As a result, adversarial features show weak alignment to any specific text prototype, leading to **both low confidence and low accuracy**. The reduced confidence shrinks the confidence–accuracy gap, producing better ECE, but the decision itself often remains incorrect.
>
> **Why our method balances both.** Our method aligns *logits* through a Dirichlet formulation rather than aligning features. This distributional alignment preserves both **inter-class semantic structure** (driving correct zero-shot predictions) and **evidence strength** (governing confidence). Consequently, adversarial predictions remain semantically meaningful and their confidence reflects underlying semantic correctness, yielding **higher robustness accuracy and good calibration at the same time**.

---

> ### Author Response · Authors · 2025-11-28
> **Summary of Revisions for Reviewer of3Z**
>
> Dear Reviewer of3Z,
>
> Thank you for your thoughtful and constructive review. We have carefully addressed your comments, and the corresponding updates have been incorporated into the revised manuscript. The main modifications made in direct response to your suggestions are summarized below:
>
> **1. On generalizability beyond CLIP-B/32 (W1).** To examine the broader applicability of our method, we added new evaluations in **Appendix F.4**, including: results on an additional CLIP variant (CLIP-B/16), and comparisons with another contrastive vision–language model (SLIP). Comprehensive results across 16 datasets are reported, demonstrating that our uncertainty calibration method generalizes beyond a single CLIP configuration.
>
> **2. On the comparison with FARE2 (W2).** We provided a detailed explanation in our point-to-point response above, clarifying the relationship between calibration behavior and robustness outcomes. We hope these clarifications help resolve the concern.
>
> We sincerely appreciate the time and effort you devoted to reviewing our work. All responses are provided above, and the revised manuscript reflects the corresponding updates.
>
> Sincerely,
>
> *The Authors*

---

### Official Review · Reviewer_KUsB · 2025-10-30

**Soundness:** 2
**Presentation:** 3
**Contribution:** 2
**Rating:** 4
**Confidence:** 4

**Summary:**

This paper aims to enhance adversarial robustness and calibration of CLIP in the zero-shot evaluation setting.

Based on the insight that adversarial examples not only hurt model performance but also suppress predictive uncertainty,
This paper proposes to calibrate uncertainty during adversarial training.

Specifically, the Dirichlet parameterization technique is borrowed from the evidential deep learning community. Applying KL divergence loss on the Dirichlet adversarial distribution and the Dirichlet clean distribution can implicitly calibrate the predictive uncertainty meanwhile enhancing adversarial robustness.

**Strengths:**

(1) The paper writes clearly and is easy to follow.

(2) Incorporating the Dirichlet parameterization technique to adversarial training is interesting.

**Weaknesses:**

(1) Beyond CLIP, the Dirichlet parameterization is a general technique and could also be applied to traditional adversarial training on the image classification task.
    In the community of adversarial training, there are lots of existing work to improve adversarial robustness, like DKL [ref1], ACAT [ref2], and TRADES.
    Would it be possible to include experiments on a standard benchmark such as CIFAR-100 or CIFAR-10 under DKL, ACAT, and TRADES setups to demonstrate that the proposed Dirichlet uncertainty calibration generalizes beyond CLIP? Otherwise, the paper should clarify why it is inherently tied to CLIP.

[ref1] Decoupled Kullback-Leibler Divergence Loss. NeurIPS 2024.
[ref2] Efficient and effective augmentation strategy for adversarial training. NeurIPS 2022..

(2) Although calibration is the central claim, the paper does not compare against previously established uncertainty calibration methods.

(3) Regarding Fig. 3(b), what data are used to evaluate the models?

**Questions:**

See weaknesses.

---

> ### Author Response · Authors · 2025-11-21
> **Response to Reviewer KUsB [1/4] — W1 (Part 1/2)**
>
> Dear Reviewer KUsB, we appreciate your thorough review and the constructive questions you raised. We address your comments point by point below and will refine the manuscript accordingly in the updated version.
>
> ---
>
> ## W1 (Part 1/2)
>
> We appreciate the reviewer’s valuable suggestion. Our method **cannot be applied** to CIFAR-10/100 under DKL, ACAT, or TRADES because the **task design motivation** is fundamentally different: supervised adversarial training focuses on *closed-set classification robustness*, whereas our Dirichlet formulation is designed for *zero-shot adversarial robustness* (ZSAR), which requires preserving the pre-trained model’s ability to generalize to **unseen classes**. Our method applies to **contrastive vision–language models (VLMs)** such as CLIP and SLIP, i.e., models trained with normalized InfoNCE logits. To make this distinction clear, we provide a structured comparison below.
>
> | Aspect                                    | Supervised AT (TRADES / ACAT / DKL)                          | UCAT (ZSAR, contrastive VLMs)                                |
> | ----------------------------------------- | ------------------------------------------------------------ | ------------------------------------------------------------ |
> | **Logit form**                            | $\ell_k(x)=g_k(f_\theta(x))$(unconstrained)                  | $\ell_k(x)={\langle v(x), t_k\rangle}/{\tau}$, where $v(x)=f_\theta(x)/\|f_\theta(x)\|$ and $t_k$ are **unit-norm** |
> | **Magnitude meaning**                     | None                                                         | Yes (magnitude encodes evidence strength)                    |
> | **Semantic geometry**                     | Not preserved (closed-set)                                   | Must preserve open-vocabulary geometry                       |
> | **Alignment target**                      | Probability distribution alignment $\mathrm{KL}(p_\text{adv}\|\|p)$ | Dirichlet distribution alignment $\mathrm{KL}(\text{Dir}(\alpha_\text{adv})\|\|\text{Dir}(\alpha))$ over semantic structure + evidence strength |
> | **Compatibility with Dirichlet evidence** | ✗ (logits have no stable or interpretable scale)             | ✓ (dirichlet formation of logits yield meaningful evidence and geometry) |
>
> **Notations:** $f_\theta$ is the image encoder; $v(x)=f_\theta(x)/\|f_\theta(x)\| $ is its normalized embedding; $p=\mathrm{softmax}(\ell)$ and $p_{\mathrm{adv}}$ are clean/adv probability distributions; $\alpha=\exp(\ell)$ and $\alpha_{\mathrm{adv}}$ denote clean/adv Dirichlet concentration parameters (Eq. 5); “Dir” is the Dirichlet distribution; “KL” denotes Kullback–Leibler divergence.
>
> The detailed technical discussion regarding why these properties make UCAT incompatible with supervised adversarial training continues in **Part 2/2** below.

---

> ### Author Response · Authors · 2025-11-21
> **Response to Reviewer KUsB [2/4] — W1 (Part 2/2)**
>
> ### W1 (Part 2/2)
>
> (This section continues the analysis from Part 1/2, following the comparison table above.)
>
> **(A) Supervised adversarial training produces logits with no meaningful or stable evidence strength**. Although supervised adversarial training (AT) optimizes logits $\ell_k(x)=g_k(f_\theta(x)$, their **absolute magnitude carries no calibrated meaning**. Softmax depends only on the *relative differences* between logits: $p_k={e^{\ell_k}}/{\text{sum}_j e^{\ell_j}}$. A global **shift** of the logits $\ell'_k=\ell_k+c$ leaves the entire distribution unchanged, and a global **rescaling** $\ell'_k=s\ell_k$ merely changes the *sharpness* of the probability simplex (analogous to modifying temperature), **without introducing a meaningful notion of evidence strength**. Thus, very different logit norms can correspond to similar probabilistic behavior, and $\|\ell(x)\|$ cannot be interpreted as calibrated confidence. Consequently, mapping supervised logits to Dirichlet concentration parameters via $\alpha_k=\exp(\ell_k)$ does not yield a semantically meaningful $\alpha_0=\text{sum}_k\alpha_k$, because its value is dominated by arbitrary logit scaling rather than genuine predictive evidence. In contrast, CLIP-style metric-similarity logits $\langle v(x), t_k\rangle/\tau$have a **geometrically constrained scale** induced by L2 normalization and a fixed temperature, making their magnitude meaningfully reflect evidence. This property is essential for our Dirichlet formulation to be well-defined and interpretable.
>
> **(B) ZSAR requires preserving open-vocabulary semantic geometry, and our Dirichlet alignment is designed specifically for this objective, whereas supervised AT neither possesses nor requires such structure.** In supervised AT, prediction is defined over a **closed label set**, and adversarial alignment operates strictly within this fixed space. The resulting geometry is inherently **task-specific** and has no mechanism to represent relationships to labels outside the training set.
>
> Zero-Shot Adversarial Robustness (ZSAR) is fundamentally different. We start from a foundation model that already exhibits strong zero-shot performance, and whose **clean outputs encode an open-vocabulary global semantic geometry**. That is, similarity relations between the image embedding and a potentially large set of text embeddings, including categories not present in the fine-tuning data. Our goal is to obtain an adversarially robust model **while intentionally preserving** this reference geometry. To achieve this, our Dirichlet alignment matches the clean output distribution of the frozen foundation model. The Dirichlet KL preserves the **class-wise concentration geometry** (capturing relative semantic structure across the vocabulary), and the **total concentration** (representing evidence strength), ensuring that adversarial perturbations do not distort the open-vocabulary similarity structure defined by the reference model.
>
> Closed-set supervised AT lacks such open-vocabulary geometry, and therefore introducing our alignment into standard AT would impose additional constraints irrelevant to its objectives.
>
> **(C) Our Dirichlet alignment is meaningful for contrastive VLM logits, not for supervised classifiers.** Our Dirichlet formulation (Sec. 4) relies on structural properties unique to **contrastive VLMs**: their logits are **bounded, L2-normalized cosine similarities** $\ell_k=\langle v(x),t_k\rangle/\tau$, which endow each logit with a **stable geometric scale** and a **shared open-vocabulary semantic coordinate system**. Under this structure, exponentiating logits yields **valid and interpretable Dirichlet evidence** (Lemma 4.2), and the resulting Dirichlet expectation is **exactly equivalent** to CLIP’s softmax prediction (Lemma 4.3), making distribution-level alignment both mathematically justified and semantically meaningful. In contrast, supervised classifiers use **unconstrained linear heads** with **unbounded** and **non-geometric** logits whose magnitude carries no evidence meaning and whose class relations are purely task-specific. Mapping such logits to Dirichlet parameters therefore produces arbitrary, non-interpretable “evidence”, and enforcing Dirichlet-KL alignment would impose a geometry that the model does not possess, conflicting with the goals of TRADES/ACAT/DKL rather than improving robustness.
>
> In summary, **our Dirichlet calibration is incompatible with supervised AT on CIFAR-10/100** because those models do not provide the normalized logits needed for meaningful Dirichlet evidence or semantic alignment. The method is instead applicable to **contrastive VLMs** (e.g., CLIP, SLIP), where such structural properties are guaranteed. Moreover, as noted in our response to **R4-of3Z (W1)**, we have already provided additional results demonstrating that our formulation extends beyond CLIP to other contrastive models. We will clarify the applicability scope in the revised manuscript.

---

> ### Author Response · Authors · 2025-11-21
> **Response to Reviewer KUsB [3/4]**
>
> ## W2
>
> We thank the reviewer for raising this point. Broadly speaking, uncertainty calibration has been extensively studied on clean data, but **very few methods address uncertainty under adversarial perturbations**, and none are directly applicable to **zero-shot adversarial robustness (ZSAR)**. Below we clarify why classical calibration methods are not meaningful baselines in our setting, and why we compare with the two representative adversarial-uncertainty approaches, Prior Networks and CCAT.
>
> ### **(1) Clean-data post-hoc UC methods do not apply to our setting**
>
> Methods developed for clean-data calibration operate in a **post-hoc** manner: they adjust predicted confidence using a **labeled validation set** to align confidence with empirical accuracy, without modifying the model’s representation or training objective. Such approaches assume: a **closed-set classifier** with fixed class logits, and **validation labels** for fitting the calibration parameters.
>
> In our setting, CLIP predictions are produced via **image–text similarity**, not a learned classifier head, and zero-shot evaluation provides **no labeled validation data** for post-hoc calibration. Moreover, because post-hoc procedures only rescale logits and do not interact with adversarial training, they cannot address the **incorrect predictions** caused by adversarial perturbations. For these structural reasons (model form + supervision dependence + training stage), classical post-hoc UC methods do not constitute meaningful baselines for adversarial zero-shot classification.
>
> ### **(2) UC methods used in adversarial detection (Prior Networks, CCAT)**
>
> A small number of works incorporate uncertainty into adversarial training, but they are specifically designed for **adversarial/OOD detection**, not robust prediction.
>
> - **Prior Networks [1]** enforce **low-precision Dirichlet targets** on adversarial examples to increase epistemic uncertainty and enable rejection.
> - **CCAT [2]** mixes one-hot and uniform targets to reduce confidence under perturbations, also aiming to detect rather than correctly classify adversarial inputs.
>
> These objectives are fundamentally different from ours: they aim to **identify adversarial inputs**, not to **classify them correctly**. Zero-shot adversarial robustness, however, requires *prediction for every input*, and no rejection mechanism is permitted.
>
> To ensure a fair comparison, we adapted Prior Networks and CCAT to the CLIP-based ZSAR setting by directly applying their original losses to the normalized image–text similarity logits, preserving all hyperparameters and uncertainty targets without CLIP-specific tuning. All models were trained on TinyImageNet with 2-step PGD and evaluated under AutoAttack ($\epsilon=1/255$), averaged over 16 datasets. Under this faithful adaptation, both methods collapse to near-zero adversarial accuracy, consistent with their design focus on uncertainty inflation rather than maintaining robust classification.
>
> ### **(3) Experimental evidence further confirms non-applicability**
>
> | Method          | Clean     | AutoAttack | Harmonic  |
> | --------------- | --------- | ---------- | --------- |
> | Prior Networks  | 27.85     | 0.90       | 1.75      |
> | CCAT            | **62.13** | 0.90       | 1.78      |
> | **Ours (UCAT)** | 54.17     | **30.58**  | **39.09** |
>
> These results highlight a **structural mismatch**: prior UC approaches purposely raise uncertainty for perturbed inputs, whereas adversarial zero-shot classification demands **accurate and calibrated predictions**. This mismatch also underscores why post-hoc calibration (which adjusts confidence but not representations) cannot address the problem.
>
> ### **(4) UCAT fills a missing methodological gap**
>
> In contrast, UCAT neither imposes uncertainty targets nor relies on post-hoc rescaling. Instead, it aligns the **full Dirichlet distributions** of clean and adversarial CLIP logits, preserving the *semantic relations*and *evidence strength* that underpin zero-shot prediction. This enables **accurate and calibrated predictions for all adversarial examples**, filling the gap left by: clean-data UC (confidence-only, no adversarial robustness) and adversarial UC (detection-only, no correct classification). The quantitative effect of UCAT is shown in Figure 1b: compared to vanilla CLIP,  our fine-tuned model achieves calibrated uncertainty levels, restoring a consistent ordering:
>
> original CLIP w/ clean img. < fine-tuned CLIP w/ clean img. < fine-tuned CLIP w/ adversarial img.,
>
> which faithfully reflects increasing input difficulty.
>
> [1] "Reverse KL-Divergence Training of Prior Networks", NeurIPS 2019.
>
> [2] "Confidence-Calibrated Adversarial Training", ICML 2020.

---

> ### Author Response · Authors · 2025-11-21
> **Response to Reviewer KUsB [4/4]**
>
> ## W3
>
> Figure 3(b) uses the averaged results over all 16 zero-shot evaluation datasets, with models trained under PGD-10 ($\epsilon = 2/255$) and evaluated using AutoAttack. We will clarify this in the revised figure caption.

---

> ### Author Response · Authors · 2025-11-28
> **Summary of Revisions for Reviewer KUsB**
>
> Dear Reviewer KUsB,
>
> We sincerely thank you for your thoughtful and constructive review. Your comments helped us clarify the scope of our method and improve the completeness of our manuscript. In the revised version, we have carefully addressed all your questions, and the corresponding modifications have been incorporated into both the main text and the appendix. Below we summarize the updates made in direct response to your suggestions:
>
> **1. Additional discussion of adversarial training methods and comparisons under ZSAR (W1).** We expanded **Appendix B.2** and added new experiments in **Appendix F.5** to directly compare our method with representative supervised adversarial training methods (TRADES, ACAT, DKL). Following their loss formulations and hyperparameters, we fine-tune each method on TinyImageNet using the same 2-step PGD ($\epsilon=1/255$) configuration as our method. All models are then evaluated zero-shot on the 16 ZSAR datasets.
>
> **2. Clarification regarding uncertainty calibration baselines (W2).** We expanded the discussion in **Appendix B.2** to explain why prior uncertainty calibration methods developed for adversarial detection are not directly applicable to zero-shot VLMs.
>
> **3. Clarification of the data used in Fig. 3(b) (W3).** We revised the caption of **Figure 3** in the main text to explicitly state that all reported numbers are averages over 16 datasets.
>
> We greatly appreciate the time and effort you invested in reviewing our work. Your comments directly improved the clarity and rigor of the manuscript. All point-to-point responses have been provided above, and the revised manuscript reflects all corresponding updates.
>
> Thank you again for your valuable feedback.
>
>
> Sincerely,
>
> *The Authors*

---

### Official Review · Reviewer_hieo · 2025-10-31

**Soundness:** 2
**Presentation:** 3
**Contribution:** 2
**Rating:** 2
**Confidence:** 4

**Summary:**

1.The paper identifies that CLIP becomes overconfident under adversarial perturbations, leading to miscalibrated uncertainty and unreliable predictions.

2.It introduces a Dirichlet-based reformulation of CLIP’s logits, enabling a unified representation of inter-class relationships and predictive confidence.

3.The authors propose UCAT, an uncertainty-calibrated adversarial fine-tuning objective that aligns full Dirichlet distributions between clean and adversarial examples rather than single-class logits.

4.Experimental results across several zero-shot benchmarks and MS-COCO demonstrate that the proposed method restores calibrated uncertainty, maintains clean accuracy, and achieves competitive adversarial robustness.

**Strengths:**

1.The paper introduces a Dirichlet-based reformulation of CLIP’s logits, which provides a theoretically grounded way to capture both inter-class relationships and predictive confidence.

2.The theoretical analysis and derivations are presented clearly and are easy to follow, making the methodology accessible to readers.

3.The paper is well-structured, with a logical flow from motivation to method to experiments, which helps communicate the ideas effectively.

4.The experiments are extensive, covering multiple zero-shot benchmarks and MS-COCO, demonstrating both the practical applicability and robustness of the approach.

**Weaknesses:**

1.The paper is motivated by the observation that CLIP can produce overconfident predictions under adversarial attacks, revealing a gap between accuracy and predictive uncertainty. However, this motivation is not sufficiently novel to fully justify the proposed solution.

2.The paper focuses on calibrating uncertainty for zero-shot adversarial CLIP, but it does not clearly explain why the proposed method is specific to CLIP or zero-shot learning. It appears that similar results could be achieved on standard image classification tasks, which raises questions about the task-specific significance of the approach.

3.The proposed method is only evaluated on CLIP. Although the theoretical analysis explains how the Dirichlet reformulation relates to traditional softmax logits and its applicability to CLIP, the paper does not clarify why CLIP is chosen for this method or what specific benefits arise from applying it to CLIP compared to other models.

4.In the ablation study, the adversarial attacks used for comparison are outdated, which limits the ability to fully demonstrate the advantages of the proposed method.

5.The analyses in Sections 3 and 4 mainly demonstrate the validity of applying the Dirichlet reformulation, but the author do not clearly show the advantages of the proposed method. The paper lacks discussion or evidence on how it outperforms existing approaches or what specific benefits it provides compared to other methods.

**Questions:**

1.The authors should clarify the connection between their method and CLIP as well as zero-shot learning. It would be helpful to explain why calibrating uncertainty is particularly meaningful in the context of zero-shot CLIP, and whether the proposed approach is specific to this setting. If the method can also be applied to other tasks, the authors could include additional experiments or comparisons to demonstrate its broader applicability and to justify the claimed contributions.

2.I hope the author could include comparisons with other vision–language models or adversarially trained models.This would help clarify whether the proposed method offers unique advantages for CLIP or if similar benefits can be achieved on other models.

3.I hope the author could include more recent and stronger attack methods to provide a more convincing evaluation of their approach.

4.The authors should clarify why using a Dirichlet distribution leads to improvements and what advantages it offers over the traditional softmax approach. They should also explain why this particular distribution is necessary for aligning the distributions of clean and adversarial samples, and  I am curious whether other distributions could be used here instead.

---

> ### Author Response · Authors · 2025-11-21
> **Response to Reviewer hieo [1/3]**
>
> Dear Reviewer hieo, thank you for carefully evaluating our work and for raising several important concerns. We provide detailed responses to each point below. The revised version of the paper will further clarify the relevant parts and address the issues you highlighted.
>
> ---
>
> ## Q1 & Q2
>
> We thank the reviewer for the insightful questions. We clarify below the relationship between our method and zero-shot CLIP, and the generality of the proposed uncertainty calibration framework.
>
> > ### **(1) Why uncertainty calibration is particularly meaningful for zero-shot CLIP**
>
> Uncertainty calibration is particularly meaningful for zero-shot CLIP because our Dirichlet formulation offers an *exact, closed-form representation* of the two quantities that zero-shot prediction fundamentally relies on. These are: (i) the **semantic structure** encoded by relative similarities to class prompts, and (ii) the **evidence strength** reflecting how strongly an image matches the entire open-vocabulary space learned during pre-training.
>
> First, our Dirichlet parameters are a strictly monotone reparameterization of CLIP’s similarity logits (Lemma 4.2), and their expectation exactly recovers CLIP’s predictive distribution (Lemma 4.3). This guarantees that **the clean Dirichlet distribution faithfully represents CLIP’s uncertainty structure**, without requiring labels from any downstream dataset.
>
> Second, because zero-shot classification selects labels solely through these relative similarities and relies on their global magnitude to gauge confidence, restoring the clean Dirichlet structure **reinstates the pre-trained open-vocabulary geometry that zero-shot recognition depends on**, even when the evaluation classes are entirely unseen.
>
> Consequently, **uncertainty calibration is particularly meaningful in the zero-shot setting because it acts directly on the core quantities that govern zero-shot decisions** and preserves the global similarity structure required for robust recognition across arbitrary, previously unseen class sets.
>
> > ### **(2) Whether our method is specific to this setting, and whether the approach transfers to other VLMs and adversarially trained models**
>
> **Our method is not specific to CLIP.** Although Sec. 3.1 presents the formulation using the standard CLIP notation, this is purely a notational choice for clarity. The derivation itself is based on the general symmetric InfoNCE objective in Eq. (1), which only requires (i) normalized embeddings on the hypersphere, (ii) a temperature-scaled dot-product similarity, and (iii) the softmax normalization used in contrastive learning. Under these conditions, the Dirichlet construction in Eq. (8) remains valid and yields a well-defined uncertainty representation for **any contrastively trained VLM**, independent of CLIP’s architecture or modality design.
>
> To verify this generality, we additionally evaluate UCAT (our Uncertainty-Calibrated Adversarial fine-Tuning method) on **CLIP-B/16** and **SLIP-B/16** (InfoNCE + SimCLR). The models are trained on TinyImageNet with 2-step PGD and evaluated under AutoAttack with $\epsilon = 1/255$, and all results are averaged over 16 datasets. Under the same training and evaluation protocol as Table 2, UCAT consistently improves AutoAttack robustness and harmonic accuracy.  These results confirm that the method extends beyond CLIP-B/32 and is applicable to a broader family of **InfoNCE-based vision–language models**.
>
> | Model      | Clean     | AutoAttack | Harmonic  |
> | ---------- | --------- | ---------- | --------- |
> | CLIP-B/16  | 63.72     | 0.01       | 0.03      |
> | **+ UCAT** | **52.91** | **30.54**  | **38.73** |
> | SLIP-B/16  | 46.03     | 0.02       | 0.05      |
> | **+ UCAT** | **38.37** | **20.40**  | **26.64** |
> | CLIP-B/32  | 64.42     | 0.51       | 1.01      |
> | **+ UCAT** | **54.17** | **30.58**  | **39.09** |

---

> ### Author Response · Authors · 2025-11-21
> **Response to Reviewer hieo [2/3]**
>
> ## Q3
>
> We thank the reviewer for this valuable suggestion. Our main evaluation already includes **PGD**, **CW**, and **AutoAttack**, which constitute the *standard and widely adopted* suite of attacks in adversarial robustness research and in all prior ZSAR works. AutoAttack (Croce & Hein, 2020) includes APGD-CE and APGD-DLR (100 iterations each) and is widely regarded as one of the most reliable and comprehensive benchmarks. These attacks are also the default evaluation protocol in recent zero-shot adversarial CLIP works such as TeCoA, FARE, and TGA-ZSR.
>
> To further strengthen our evaluation, we additionally tested our UCAT under two **more recent and competitive** attacks:
>
> * **CAA** (AAAI 2021 [1]), a strong composite adversarial attack;
> * **A$^3$** (CVPR 2022 [2]), the adaptive AutoAttack variant and the **1st-place method** among 1681 teams in the CVPR 2021 “White-box Adversarial Attacks on Defense Models” challenge.
>
> Under the same protocol as Table 4 (Trained on TinyImageNet with 10-step PGD and evaluated under AutoAttack with $\epsilon = 2/255$), and **averaged over the same 16 evaluation datasets**, our method demonstrates the strongest robustness and best clean–robustness trade-off:
>
> | Method                                 | Clean     | CAA       | Harmonic  |
> | -------------------------------------- | --------- | --------- | --------- |
> | CLIP (Backbone, w/o adversarial fine-tuning) | 64.46     | 5.32      | 9.83      |
> | TeCoA                                  | 36.72     | 20.59     | 26.38     |
> | FARE                                   | 21.41     | 12.99     | 16.17     |
> | PMG-AFT                                | **53.32** | 11.54     | 18.97     |
> | TGA-ZSR                                | 53.25     | 14.72     | 23.06     |
> | **UCAT (ours)**                        | 45.21     | **22.21** | **29.79** |
>
> | Method                                 | Clean     | A$^3$     | Harmonic  |
> | -------------------------------------- | --------- | --------- | --------- |
> | CLIP (Backbone, w/o adversarial fine-tuning) | 64.46     | 5.58      | 10.28     |
> | TeCoA                                  | 36.72     | 20.52     | 26.32     |
> | FARE                                   | 21.41     | 12.95     | 16.14     |
> | PMG-AFT                                | **53.32** | 11.83     | 19.36     |
> | TGA-ZSR                                | 53.25     | 19.64     | 28.70     |
> | **UCAT (ours)**                        | 45.21     | **22.09** | **29.68** |
>
> [1] "Composite Adversarial Attacks", AAAI 2021.
>
> [2] "Practical Evaluation of Adversarial Robustness via Adaptive Auto Attack", CVPR 2022.

---

> ### Author Response · Authors · 2025-11-21
> **Response to Reviewer hieo [3/3]**
>
> ## Q4
>
> Below we provide a structured clarification in four parts: (1) why a Dirichlet reformulation is necessary; (2) how it leads to improvements; (3) what advantages it offers over the traditional softmax view; and (4) whether alternative distributions could serve the same role.
>
> > ### **(1) Why Dirichlet is necessary?**
>
> **(a)** **Dirichlet is not an extra assumption but a structural reinterpretation of the contrastive logits.** In Sec. 4, we define concentration parameters directly from the similarity logits (Eq. 8). Lemma 4.2 shows that this mapping is strictly monotone, so **the full semantic geometry of the logits is preserved**. Lemma 4.3 shows that the expectation of this Dirichlet distribution matches CLIP’s predictive behavior exactly when $\tau'=\tau$. Thus, the Dirichlet family faithfully retains the information CLIP uses for zero-shot prediction.
>
> **(b)** **Dirichlet exposes an additional quantity crucial for zero-shot robustness: evidence strength.** Under the uncertainty decomposition in Sec. 3.3: relative concentration captures **semantic relations** across prompts, while total concentration $\alpha_0$ reflects **how strongly the image matches the entire pre-trained open-vocabulary space**. This global similarity strength is part of CLIP’s pre-training signal and is essential for generalizing to unseen classes. Dirichlet is particularly suitable in *our formulation*, because it provides closed-form access to semantic structure + evidence strength, and is directly derived from contrastive logits.
>
> > ### **(2) Why Dirichlet leads to improvements？**
>
> **(a) The clean Dirichlet distribution comes from the frozen model and therefore captures the open-world semantic structure learned during pre-training.** It reflects CLIP’s broad understanding of category relationships, far beyond the limited labels of the current training split.
>
> **(b) Aligning adversarial predictions to this clean distribution forces the fine-tuned model to retain both:** **semantic geometry** that discriminates among any prompt set, and **evidence strength** that indicates how confidently the model situates an image within the open-vocabulary space.
>
> **(c) These two components are exactly the factors that zero-shot recognition depends on.** Since evaluation classes are unseen, robustness does not come from memorizing specific labels but from **preserving the pre-trained open-vocabulary structure**. Dirichlet alignment improves robustness precisely because it aligns the quantities that drive zero-shot decisions.
>
> > ### **(3) What advantages it offers over the traditional softmax approach**
>
> The key advantage of Dirichlet over softmax is that it preserves **evidence strength** (the absolute magnitude of the similarity logits) while softmax removes it, and this quantity is essential for zero-shot adversarial robustness (ZSAR). Softmax removes logit magnitude because its normalization divides by the sum of exponentiated logits, forcing the output to depend only on relative differences. In contrast, the Dirichlet concentration vector retains magnitude explicitly through its total mass $\alpha_0$, which grows proportionally with the absolute size of the logits.
>
> This global matching magnitude is crucial for zero-shot robustness: it determines how confidently CLIP situates an image among *unseen* categories, which is exactly the setting where ZSAR must operate. Dirichlet makes evidence strength explicit through its total concentration  $\alpha_0$, allowing the alignment loss to regularize **both** semantic relations and global matching confidence. This prevents adversarial updates from collapsing the open-world similarity strength that CLIP relies on, while softmax-only alignment cannot access or preserve this information.
>
> **In summary, Dirichlet improves ZSAR because it exposes evidence strength which softmax discards, and enables clean–adversarial alignment over a signal that zero-shot recognition fundamentally depends on.**
>
> > ### **(4) Whether other distributions could be used instead**
>
> Our reformulation in Sec. 4 is **not a modeling assumption** but **follows directly from the contrastive logit structure**. The Dirichlet family is used because it offers three properties that our derivation relies on: (i) support on the probability simplex, (ii) an exponential concentration parameterization that preserves the logit ordering, and (iii) a closed-form expectation that remains aligned with the contrastive prediction rule.
>
> Common alternatives, such as Gaussian or logistic-normal distributions, do not jointly provide these properties, particularly the simplex support and the closed-form concentration interpretation required for our calibration objective. For this reason, they would not integrate consistently with the reformulation in Sec. 4 or support the alignment objective in Sec. 5.

---

> ### Author Response · Authors · 2025-11-28
> **Summary of Revisions for Reviewer hieo**
>
> Dear Reviewer hieo,
>
> Thank you for your detailed review and for raising several important points. Your feedback helped us further strengthen the scope and rigor of our study. In revising the manuscript, we systematically addressed each of your concerns, and the corresponding updates are now fully incorporated into the revised version. Below we highlight the main changes made in direct response to your suggestions:
>
> **1. Applicability beyond CLIP and additional VLM experiments (Q1 & Q2).** To examine whether our method generalizes beyond the specific CLIP-B/32 model, we have added new experiments in **Appendix F.4**, including: results on another CLIP variant (CLIP-B/16), and comparisons with an additional contrastive vision–language model (SLIP). The detailed results across 16 datasets demonstrate that the proposed uncertainty calibration consistently improves zero-shot adversarial robustness across these models.
>
> **2. Inclusion of stronger adversarial attacks (Q3).** Following your suggestion, we expanded our evaluation to include two widely adopted and stronger attack methods released after autoattack, CAA and A$^3$, which have been validated by prior works as reliable robustness benchmarks. The complete results across 16 datasets are provided in **Appendix F.2**, further supporting the effectiveness of our approach under stronger threat models.
>
> We sincerely appreciate the time and effort you devoted to reviewing our work. All point-to-point responses have been provided above, and the revised manuscript has been updated accordingly.
>
> Thank you for your constructive feedback.
>
> Sincerely,
>
> *The Authors*

---

### Official Review · Reviewer_f5Be · 2025-11-01

**Soundness:** 3
**Presentation:** 3
**Contribution:** 3
**Rating:** 6
**Confidence:** 4

**Summary:**

This paper proposes to enhance the zero-shot classification robustness of the CLIP model, an Uncertainty-Calibrated Adversarial fine-Tuning framework for CLIP (UCAT). The authors introduce a loss function for adversarial training that considers both prediction accuracy and uncertainty alignments. The method is evaluated on multiple datasets and shows improved robustness against adversarial attacks compared to existing methods.

**Strengths:**

The motivation for improving CLIP's zero-shot robustness is well articulated, and the proposed method is supported by thorough experiments. The authors provide comprehensive evaluations on various datasets and attack methods, demonstrating the effectiveness of their approach. The ablation studies further validate the contributions of different components of the proposed loss function.

**Weaknesses:**

1. The choice of concentration parameter alpha for the Dirichlet distribution in Definition 4.1 is not well justified. The authors should provide insights into how this parameter is chosen and its sensitivity to performance.

2. The proposed method shows lower performance on certain datasets (SUN397 and PCAM) as seen in Table 1. The authors should discuss potential reasons for this discrepancy.

See the questions.

**Questions:**

1. Since the goal of this paper is to improve the zero-shot robustness of CLIP, when the labels of the downstream tasks are unknown during training, how does the loss term encourage uncertainty alignment? Is there any insight into why this helps improve robustness on those tasks?

2. Since the Dirichlet distribution is a Bayesian prior over categorical distributions, it is usually sensitive to the choice of concentration parameters. Definition 4.1 introduces a concentration parameter alpha for the Dirichlet distribution. Do the authors have any insights on why they chose this parameter and how sensitive the performance is to this choice? Also, parameters such as tau prime should be considered in ablation studies.

3. The proposed methods outperform existing methods on most datasets, but there are some cases where the performance is consistently lower (SUN397 and PCAM in Table 1). Do the authors have any insights into why this happens?

4. In Table 2 and Table 5, why is the clean average accuracy not consistent？ Since they are all evaluated on the same datasets, shouldn't the clean accuracies be identical?

---

> ### Author Response · Authors · 2025-11-21
> **Response to Reviewer f5Be [1/3]**
>
> Dear Reviewer f5Be, thank you sincerely for your constructive and encouraging feedback. We greatly appreciate your time and insights. Below we respond to each of your comments and clarify the technical points raised. We will incorporate the corresponding improvements into the revised version of the paper.
>
> ---
>
> ## Q1
>
> Thank you for the question. Below we address the reviewer’s question in two layers—**(1) how the loss encourages uncertainty alignment without downstream labels**, and **(2) why such alignment naturally leads to improved zero-shot robustness**.
>
> > ### **(1) How the loss encourages uncertainty alignment without downstream labels**
>
> During fine-tuning, we only use labels from the adversarial training data, but our zero-shot evaluation tasks naturally have *no labels*. Our uncertainty calibration loss circumvents the need for any downstream labels by enforcing **distributional alignment** between:
>
> * the **clean Dirichlet distribution** produced by the frozen CLIP model, and
>
> * the **adversarial Dirichlet distribution** produced by the fine-tuned CLIP model.
>
> Under our Dirichlet reformulation (Sec. 4), clean logits are mapped to concentration parameters $\alpha$ that encode
>
> 1. **Inter-class semantic relations** — captured by the *relative geometry* of the logits.
>
> 2. **Evidence strength** — captured by the total concentration $\alpha_0$.
>
> These two components correspond exactly to the aleatoric/epistemic uncertainty decomposition in Sec. 3.3. Adversarial perturbations distort both aspects. Our uncertainty calibration regularization term
> $$
> L_{\text{ucr}}=\mathrm{KL}(\mathrm{Dir}(\alpha_{\text{adv}})\|\mathrm{Dir}(\alpha))
> $$
> aligns the full distributions. It restores semantic structure and calibrated evidence of the clean CLIP model in a single step, even for classes unseen during training. This aligns uncertainty in a **task-agnostic** manner. The target comes fully from CLIP itself rather than any downstream supervision.
>
> > ### **(2) Why this improves zero-shot robustness without downstream labels**
>
> Zero-shot prediction depends purely on CLIP’s **global image–text similarity geometry**, not on task-specific labels. The clean Dirichlet distribution captures exactly this geometry: it embeds how an image relates to the **entire vocabulary of text prototypes**, not any particular class list. Because the geometry is:
>
> * global, task-agnostic, and fixed for any unseen dataset,
>
> forcing adversarial predictions to match clean Dirichlet distributions preserves:
>
> * **semantic fidelity** (inter-class structure), **calibrated confidence** (evidence strength), **robust similarity alignment** to the text space.
>
> Thus, even though downstream tasks do not provide labels, preserving CLIP’s intrinsic uncertainty structure ensures that adversarial perturbations cannot distort the text–image alignment that zero-shot recognition fundamentally relies on.
>
> **In summary**, zero-shot robustness improves **without downstream labels** because the loss aligns adversarial predictions to clean CLIP-derived Dirichlet distributions, which encode task-agnostic semantic structure and uncertainty. Besides, preserving this structure directly supports zero-shot recognition on unseen class sets. Therefore, **the entire robustness gain comes from distributional uncertainty alignment derived from CLIP itself, not from any downstream supervision**.

---

> ### Author Response · Authors · 2025-11-21
> **Response to Reviewer f5Be [2/3]**
>
> ## Q2
>
> Thank you for pointing this out. We would like to clarify a key distinction:
>
> **the concentration parameters $\alpha$ in Definition 4.1 are *not* manually chosen hyperparameters**.
>
> They are **the network outputs**, predicted from CLIP logits via our Dirichlet reformulation. Thus, the model learns $\alpha$ directly from data, and no fixed $\alpha$ needs to be selected or tuned.
>
> The only manually specified scalar is the **calibration coefficient $\tau^\prime$**, which serves as the temperature in the logit-to-evidence mapping. This parameter is unrelated to Dirichlet concentration and follows standard practice in contrastive learning. The value **$\tau^\prime = 0.07$** is widely used in MoCo[1], CLIP, and ALIGN[2], and prior work [3,4,5] shows it yields stable representation geometry.
>
> To address the reviewer’s concern about sensitivity, we performed an ablation over $\tau^\prime$ (same protocol as Table 2). Trained on TinyImageNet with 2-step PGD and evaluated under AutoAttack with $\epsilon = 1/255$. Results are averaged over 16 datasets.
>
> | τ′       | Clean     | AutoAttack | Harmonic  |
> | -------- | --------- | ---------- | --------- |
> | 0.01     | 44.98     | 28.89      | 35.18     |
> | 0.03     | 44.01     | 28.50      | 34.60     |
> | 0.05     | 46.65     | 29.77      | 36.35     |
> | **0.07** | **54.17** | **30.58**  | **39.09** |
> | 0.09     | 55.25     | 28.94      | 37.99     |
> | 0.10     | 55.18     | 27.88      | 37.05     |
>
> The reported results show that **0.07 consistently achieves the best trade-off.** This clarification and ablation of the complete 16-dataset results will be included in the revision.
>
> [1] "Momentum contrast for unsupervised visual representation learning", CVPR 2020.
>
> [2] "Scaling Up Visual and Vision-Language Representation Learning With Noisy Text Supervision", ICML 2021.
>
> [3] "Understanding Contrastive Representation Learning through Alignment and Uniformity on the Hypersphere", ICML 2020.
>
> [4] "Understanding the Behaviour of Contrastive Loss", CVPR 2021.
>
> [5] "Decoupled Contrastive Learning", ECCV 2022.

---

> ### Author Response · Authors · 2025-11-21
> **Response to Reviewer f5Be [3/3]**
>
> ## Q3
>
> We thank the reviewer for the question. Below we summarize the empirical trends, explain the underlying reasons, and clarify when **UCAT (our Uncertainty-Calibrated Adversarial fine-Tuning method)** is expected to be most effective.
>
> **(1) Empirical observations.** Across the **16 datasets** in Table 2 and Table 5, UCAT achieves **the best or near-best clean–robustness performance on the vast majority of them**. This includes most **natural-image datasets**, such as **general object recognition** (ImageNet, CIFAR-10/100), **scene recognition** (SUN397), and **fine-grained categories** (Caltech-101/256, Flowers102, Food101). The datasets where the improvements are noticeably smaller are **EuroSAT** and **PCAM**, whereas SUN397 maintains strong performance under our method.
>
> **(2) Why these datasets behave differently.** UCAT improves robustness by aligning adversarial predictions with the **clean Dirichlet geometry** of the frozen CLIP model. This mechanism is particularly effective when CLIP provides a **meaningful and reliable semantic similarity structure**, which holds for most natural-image datasets close to the model’s pre-training distribution.
>
> In contrast, **EuroSAT (satellite imagery)** and **PCAM (histopathology patches)** exhibit substantial **distribution shift** away from CLIP’s natural-image domain. As reflected in Fig. 1(b), CLIP’s clean predictions on these datasets already show **higher uncertainty** and **weaker inter-class structure**. When the clean geometry is inherently less informative, there is **less robust structure for UCAT to preserve** through Dirichlet alignment, naturally limiting the magnitude of improvement. This reflects domain difficulty rather than a failure of the method.
>
> Even so, UCAT maintains a **stable and well-balanced clean–robustness harmonic mean** across clean and AutoAttack evaluations on these datasets (Tables 2 and 5), consistent with the intended behavior under both adversarial and distributional shifts.
>
> **(3) Summary When UCAT is most effective.** In summary, UCAT is **most effective** on datasets where CLIP provides reliable clean semantic structure. This accounts for the **majority of the 16 evaluated datasets**. On datasets with strong distribution shift, UCAT remains stable, though the absolute gains are naturally constrained by the limited clean geometry available from the pre-trained encoder.
>
> ---
>
> ## Q4
>
> We thank the reviewer for the question. We would like to clarify that the “clean average accuracy” reported in Table 2 and Table 5 **does not come from the original frozen CLIP model**, but from the **fine-tuned models obtained under two different adversarial training settings**. These settings follow the respective protocols of prior works we compare to.
>
> Table 2 uses the *light PGD setting* (2-step, $\epsilon = 1/255$) adopted in **TeCoA**, while Table 5 uses the *strong PGD setting* (10-step, $\epsilon = 2/255$) adopted in **FARE**. These two methods represent **the most established baselines for zero-shot adversarial robustness**, and therefore we follow their respective setups for fair comparison. Because the perturbation strengths, optimization trajectories, and adversarial signals differ, the models learned under these two settings are not identical, and thus their **clean accuracies naturally differ as well**.
>
> Within each table, the **deep-gray entries correspond to the clean accuracy of the frozen CLIP model**, evaluated consistently using CLIP’s standard protocol. In contrast, the clean accuracies of **our fine-tuned models** differ between the two tables because they originate from **two distinct fine-tuning settings**. We will clarify this distinction more explicitly in the revised manuscript.

---

> ### Author Response · Authors · 2025-11-28
> **Summary of Revisions for Reviewer f5Be**
>
> Dear Reviewer f5Be,
>
> We sincerely thank you for your thorough and constructive review. Your comments significantly improved the clarity and completeness of our manuscript. In the revised version, we have carefully addressed all your questions, and the corresponding modifications have been explicitly incorporated into the revised manuscript. Below we summarize the key updates made in direct response to your suggestions:
>
> **1. On the sensitivity to concentration parameters (Q2).** We have expanded the **Remark following Corollary 4.3.1** to explain our choice of the temperature parameter $\tau^\prime=0.07$, including its theoretical motivation, empirical justification, and consistency with prior works (MoCo, CLIP, ALIGN). In **Appendix F.3**, we additionally provide a detailed analysis across 16 datasets to demonstrate robustness with respect to $\tau^\prime$.
>
> **2. On performance variations across datasets (Q3).** In **Sec. 6.2**, we added a clearer discussion of the behavior on EuroSAT and PCAM, analyzing dataset-specific factors and clarifying the effective regime where our uncertainty calibration yields the most benefit.
>
> **3. Distinction between Table 2 and Table 5 (Q4).** We corrected **Table 5** and refined the corresponding content in **Appendix F.2** to ensure consistency with Table 2 and to clearly differentiate the evaluation settings.
>
> We appreciate your valuable feedback once again. All point-to-point responses are included in our response above, and the revised manuscript has been updated accordingly.
>
> Thank you for your time and consideration.
>
> Sincerely,
>
> *The Authors*

---

### Meta-Review · Area_Chair_vxKp · 2026-01-21

**Summary:**

This paper has received mixed reviews with scores 6, 2, 4, 8.

This work proposes to leverage the concentration parameters of Dirichlet distribution to model uncertainty. The goal is to capture correlations between classes as motivated by shortcomings of single-class anchor alignment. The reviewers had a number of concerns, including limited novelty, lack of specificity of solution to zero-shot adversarial tasks (or demonstration on wider set of problems), limited comparisons with related works. On balance, this sentiment is shared by the meta-reviewer.

Additionally, the main motivation for the paper to leverage uncertainty is to capture correlations as per "pulling features
along an unconstrained direction and disregarding the relative geometry of neighboring embeddings". However, if that is the motivation for this work, covariance type of alignment should be utilized to explicitly capture correlations, perhaps similarly to works like "Robustifying Zero-Shot Vision Language Models by Subspaces Alignment", ICCV25 which are multi-anchor alignment models. Another way to modeling correlations would be the use of an appropriate classifier.

Therefore, while the use of Dirichlet concentration parameters to model uncertainty is nice, the key assumptions of the proposed idea need to be justified better, e.g., why not to model correlations in a more explicit way, why uncertainty modeling is a better way, by how much? Moreover, related works need to be updated to reflect several zero-shot adversarial VLMs (there is a sizable body of such works but this paper's related works section in lines 104-114 mentions only four works).

Minor comment:
- "text encoder remains fixed and provides stable semantic anchors" - some zero-shot adversarial VMLs also fine-tune the text head or tunable prompt

**Reviewer Concerns:**

Reviewer f5Be's concerns about sensitivity to concentration (somewhat).

Reviewer of3Z's concerns about single ViT backbone eval.

**Reviewer Scores:**

Reviewer hieo would likely maintain score below 6 as the motivation for uncertainty remain loose even taking into account the authors' response. It is just one arbitrary choice to tackle the single-anchor problem.

Reviewer KUsB would likely maintain score below 6 as due to conviction this idea is more universal than CLIP. Indeed, label correlation and single-anchor problem is relevant also to distillation models, etc.

The other reviewers would have maintained their positive scores.

---

### Decision · Program_Chairs · 2026-01-26

Reject